# DECN: Evolution Inspired Deep Convolution Network for Black-box Optimization

## Abstract

We design a deep evolution convolution network (DECN) to overcome the poor generalization of an evolutionary algorithm in handling continuous black-box optimization. DECN is composed of two modules: convolution-based reasoning module (CRM) and selection module (SM), to move from hand-designed optimization strategies to learned optimization strategies. CRM produces a population closer to the optimal solution based on the convolution operators, and SM removes poor solutions. We also design a proper loss function to train DECN so as to force the random population to move near the optimal solution. The experimental results on one synthetic case and two real-world cases show the advantages of learned optimization strategies over human-designed black-box optimization baselines. DECN obtains good performance with deep structure but encounters difficulties in training. In addition, DECN is friendly to the acceleration with Graphics Processing Units (GPUs) and runs 102 times faster than unaccelerated EA when evolving 32 populations, each containing 6400 individuals.

## 1 Introduction

Optimization has been an old and essential research topic in history; Many tasks in computer vision, machine learning, and natural language processing can be abstracted as optimization problems. Moreover, many of these problems are black-box, such as neural architecture search (Elsken et al., 2019) and hyperparameter optimization (Hutter et al., 2019). Various approaches, such as Bayesian optimization (Snoek et al., 2012) and evolutionary algorithms (EAs), including genetic algorithms (Jin et al., 2019; Khadka & Tumer, 2018; Zhang & Li, 2007; Such et al., 2017; Stanley et al., 2019) and evolution strategies (ES) (Wierstra et al., 2014; Vicol et al., 2021; Hansen & Ostermeier, 2001; Auger & Hansen, 2005; Salimans et al., 2017), have been proposed to deal with these problems in the past.

The generalization ability of EAs is poor. Faced with a new black-box optimization task, we need experts to redesign the EA's crossover, mutation, and selection operations to maximize its performance on the target task, resulting in a hand-designed EA with big application limitation. Most importantly, due to the limitation of expert knowledge, only little target function information is used to assist the design of EA, which makes it challenging to adapt to the target task. How to automatically design optimization strategies according to new tasks is crucial. EA is a generative optimization model that realizes the generation from a random population to an optimal solution by generating potential solutions and retaining good solutions. The task of automatically designing an optimization strategy is learning how to automatically generate and retain potential solutions.

This paper first attempts to develop a deep evolution convolution network (DECN) to learn to exploit structure in the problem of interest so that DECN can automatically move a random population near the optimal solution for different black-box optimization tasks. DECN uses the process of EAs to guide the design of this new learning-to-optimize architecture. Like EAs, we propose two critical components of DECN to generate and select potential solutions: a convolution-based reasoning module (CRM) and a selection module (SM). For CRM, we need to ensure the exchange of information between individuals in the population to achieve the function of generating potential solutions. We design a lattice-like environment organizing the population into the modified convolution operators and then employ mirror padding (Goodfellow et al., 2016) to generate the potential offspring. SM need to update the population to survive the fittest solutions. We design SM based on a pairwise

comparison between the offspring and the input population regarding their fitness, implemented by employing the mask operator. Then, we design the evolution module (EM) based on CRM and SM to simulate one generation of EAs. Finally, we build the DECN by stacking several EMs to cope with the first issue.

The untrained DECN does not handle the black-box optimization problem well because it needs information about the target black-box function. In order to better optimize the objective task, we need to design a training set containing objective function information and a practical loss function to guide the parameter training of DECN. The characteristics of black-box functions make it difficult for us to obtain their gradient information to assist in the training of DECN. To overcome the second issue, the following questions must be solved: how to design a proper loss function and training dataset. We construct a differentiable surrogate function set of the target black-box function to obtain the information of the target black-box function. However, the optimal population is usually unknown. The designed loss function is to maximize the difference between the initial and output populations to train DECN towards the optimal solution, where the loss function can be optimized by back-propagation.

We test the performance of DECN on six standard black-box functions, protein docking problem, and planner mechanic arm problem. Three population-based optimization baselines, Bayesian optimization (Kandasamy et al., 2020), and a learning-to-optimize method for black-box optimization (Cao et al., 2019) are employed as references. The results indicate that DECN can automatically learn efficient mapping for unconstrained continuous optimization on high-fidelity and low-fidelity training datasets. Finally, to verify that DECN is friendly to Graphics Processing Units (GPUs)' acceleration, we compare the runtime of DECNs on one 1080Ti GPU with the standard EA.

## 2 RELATED WORK

There are many efforts that can handle black-box optimization, such as Bayesian optimization (Snoek et al., 2012) and EAs (Mitchell, 1998). Since the object of DECN is population, it has a strong relationship with EA. Meanwhile, DECN is a new learning-to-optimize (L2O) framework. Appendix A.10 shows our detailed motivations.

**EAs**. EAs are inspired by the evolution of species and have provided acceptable performance for black-box optimization. There are two essential parts to EAs: 1) crossover and mutation: how to generate individuals with the potential to approach the optimal solution; 2) selection: how to discard individuals with inferior performance while maintaining the ones with superior performance. In the past decades, many algorithmic components have been designed for different tasks in EAs. The performance of algorithms varies towards various tasks, as different optimization strategies may be required given diverse landscapes. This paper focuses on two critical issues of EAs: 1) Poor generalization ability. Existing methods manually adjust genetic operators' hyperparameters and design the combination between them (Kerschke et al., 2019; Tian et al., 2020); However, its crossover, mutation, and selection modules can only be designed manually based on expert knowledge and cannot effectively interact with the environment (function); that is, they cannot change their elements in real-time to adapt to new problems through the feedback of the objective function. 2) The acceleration of EAs using GPUs is a challenging task. The support for multiple subpopulations to evolve simultaneously has paramount significance in practical applications. Besides, many available genetic operators are unfriendly to the GPU acceleration, as GPUs are weak in processing logical operations. DECN overcomes the above issues. It is adapted to different optimization scenarios, based on which DECN automatically forms optimization strategies.

**L2O**. The most related work is about L2O (Chen et al., 2022). These methods employ the long short-term memory architecture (LSTM) (Chen et al., 2020; Andrychowicz et al., 2016; Chen et al., 2017; Li & Malik, 2016; Wichrowska et al., 2017; Bello et al., 2017) or multilayer perceptron (MLP) (Metz et al., 2019) as the optimizer to achieve point-based optimization (Sun et al., 2018; Vicol et al., 2021; Flennerhag et al., 2021; Li & Malik, 2016). However, none of the above methods can handle black-box optimization. Swarm-inspired meta-optimizer (Cao et al., 2019) learns in the algorithmic space of both point-based and population-based optimization algorithms. This method does not consider the advantage of EAs and is a model-free method. Existing L2O techniques rarely focus on black-box optimization. Although several efforts like (Cao et al., 2019; Chen et al., 2017) have coped with these problems, they all deal with small-scale problems in the experimental setting. DECN is a

new L2O framework that makes up for the performance disadvantage of the current L2O architecture in black-box optimization. This paper makes an essential contribution to the L2O community.

## 3   DEEP EVOLUTION CONVOLUTION NETWORK

### 3.1   PROBLEM DEFINITION

An unconstrained black-box optimization problem can be transformed or represented by a minimization problem, and constraints may exist for corresponding solutions:

$$\min \ f(s|\xi), s.t. \ x_i \in [d_i, u_i], \forall x_i \in s, \tag{1}$$

where $s = (x_1, x_2, \cdots, x_D)$ represents the solution of optimization problem $f$ while $d = (d_1, d_2, \cdots, d_D)$ and $u = (u_1, u_2, \cdots, u_D)$ denote the corresponding lower and upper bounds of the solution's domain, respectively. $\xi$ is the known parameters of $f$. We can only use the query-response terminology because the objective function $f$ is a black box without a closed-form formulation in this setting. Suppose $n$ individuals of one population $(S = \{s_1, \cdots, s_n\})$ be $s_1 = (x_1^1, x_2^1, \cdots, x_D^1), s_2 = (x_1^2, x_2^2, \cdots, x_D^2), \cdots, s_n = (x_1^n, x_2^n, \cdots, x_D^n)$. This paper aims to make the initial population move near the optimal solution. To be noted, $\theta$ is the parameters (strategies) of $G$, $G$ is an abstract function remarking the optimization process, $S_0$ is the initial population, and $S_t$ is the output population. The procedure of DECN can be formulated as $S_t = G_\theta(S_0, f(s|\xi))$. Based on the optimized $\theta$, DECN optimizes $f(s|\xi)$ by $G_\theta$.

### 3.2   CONVOLUTION-BASED REASONING MODULE

We design CRM to ensure that individuals in the population can exchange information to generate the potential solutions near the optimal solution (similar to the function of recombination operator in EAs). The corresponding correction to the convolution operator can achieve this goal, which is the motivation for our design with convolution. This part mainly describes how to construct CRM to generate new solutions.

**Organize Population into Convolution**. We arrange all individuals in a lattice-like environment with a size of $L \times L$. In this case, we can represent the population by using a tensor $(i, j, d)$, where $(i, j)$ locates the position of one individual $S(i, j)$ in the $L \times L$ lattice and $d$ is the dimension information of this individual. Appendix A.2 gives an illustration of the tensor data. The individuals in the lattice are sorted in descending order to construct a population tensor with a consistent pattern (see Figure 5 of Appendix). The number of channels in input tensors is $D+1$, where $D$ is the dimension of the optimization task, and the fitness of individuals occupies one channel. The fitness channel does not participate in the convolution process but is essential for the information selection in the selection module.

**How to Design CRM**. After organizing the population into a tensor $(L, L, D + 1)$, we modified the depthwise separable convolution (DSC) operator (Chollet, 2017) to generate new individuals by merging information in different dimensions among individuals. The DSC operator includes a depthwise convolution followed by a pointwise convolution. Pointwise convolution maps the output channel of depthwise convolution to a new channel space. When applied to our task, we remove the pointwise convolution in DSC to avoid the information interaction between channels. Eq. (2) provides the details about how to reproduce offspring, and one example is shown in Fig. 8 of the Appendix.

$$S^{'}(i, j) = \sum_{k,l} w_{k,l} S(i + k, j + l), \tag{2}$$

where $S^{'}(i, j)$ denotes the individuals in the output population, $S(i, j)$ denotes the individuals in the input population, and $w_{k,l}$ represents the related parameters of convolution kernels. Moreover, to adapt to optimization tasks with different dimensions, different channels share the same parameters. The parameters within convolution kernels record the strategies learned by this module to reason over available populations given different tasks. There are still two critical issues to address here.

1) *Since there does not exist a consistent pattern in the population, the gradient upon parameters is unstable as well as divergent.* A fitness-sensitive convolution is designed, where the CRM's attention to available information should be relative to the quality and diversity of the population. $w_{k,l}$ reflects

the module's attention during reasoning and is usually relative to the fitness of individuals. After that, this problem is resolved by simply sorting the population in the lattice based on individuals' fitness.

2) *Another vital issue is the scale of the offspring.* We conduct padding before the convolution operator to maintain the same scale as the input population. However, filling the tensor of the population with constant values '0' is not proper, as is usually done in computer vision. Instead, mirror padding copies the individuals to maintain the same scale between the offspring and the input population. As the recombination process conducts the information interaction among individuals, copying the individual is better than extending tensors with a constant value. An implementation of mirror padding to the population is given in Appendix.A.3.

The size of convolution kernels within CRM determines the number of individuals employed to implement reasoning of $S^{'}(i, j)$. After that, this paper employs convolution kernels with commonly used sizes. Different convolution kernels produce corresponding output tensors, while the final offspring are obtained by averaging multiple convolutions' output. Then, the fitness of this final offspring will be evaluated.

### 3.3 SELECTION MODULE

The selection module updates the population so as to survive the fittest individuals. SM updates individuals based on a pairwise comparison between the offspring and input population regarding their fitness for efficiency and simplicity. $S_{i-1}$ and $S^{'}_{i-1}$ are the input and output populations of CRM, respectively. $S_{i-1}$ and $S^{'}_{i-1}$ contain $D+1$ channels. The first channel stores the fitness value of an individual. Thereafter, a matrix subtraction of fitness channel corresponding to $S_{i-1}$ and $S^{'}_{i-1}$ compares the quality of individuals from $S_{i-1}$ and $S^{'}_{i-1}$ pairwise. A binary mask matrix indicating the selected individual can be obtained based on the indicator function $l_{x>0}(x)$, where $l_{x>0}(x) = 1$ if $x > 0$ and $l_{x>0}(x) = 0$ if $x < 0$. To extract selected

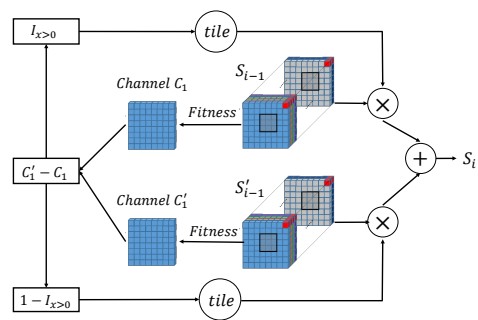

Figure 1: SM. An indication matrix is produced by subtracting the fitness channel ($c_1$), based on which individuals within the input and output populations can be extracted to form offspring.

individuals from $S_{i-1}$ and $S^{'}_{i-1}$, we construct a binary mask tensor by copying and extending the mask matrix to the same shape as $S_{i-1}$ and $S^{'}_{i-1}$. The selected information forms a new tensor $S_i$ by employing Eq. (3) illustrated in Fig. 1.

$$S_i = tile(l_{x>0}(M_{F'} - M_F)) \bullet S_{i-1} + tile(1 - l_{x>0}(M_{F'} - M_F)) \bullet S^{'}_{i-1}, \quad (3)$$

where the *tile* copy function extends the indication matrix to a tensor with size $(L, L, D)$, $M_F(M_{F'})$ denotes the fitness matrix of $S_{i-1}(S^{'}_{i-1})$, and $\bullet$ indicates the pairwise multiplication between inputs.

### 3.4 THE STRUCTURE OF DECN

In vanilla EAs, each generation consists of recombination and selection operations. Like EAs, in Fig. 2, a learnable module based on CRM and SM is designed to learn optimization strategies, termed evolution module (EM). Then, DECN is established by stacking several EMs to simulate generations within EAs. $S_{i-1}$ is the input population of $EM_i$. $S^{'}_{i-1}$ is the output of CRM in order to further improve the quality of individuals in the global and local search scopes. Then, SM

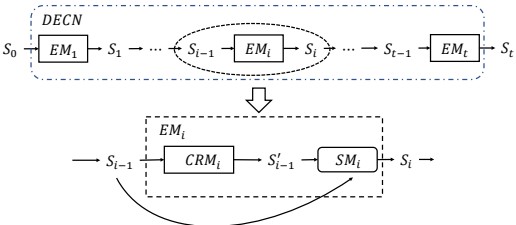

Figure 2: A general view of DECN and EM.

selects the valuable individuals from $S_{i-1}$ and $S'_{i-1}$ according to their function fitness. Fig. 3 provides an intuitive description of data flow in EM.

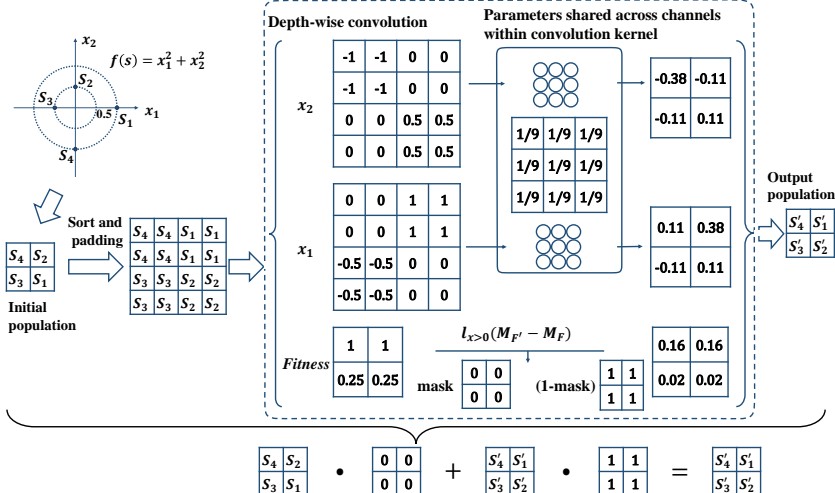

Figure 3: An example to show the data flow in EM. Suppose $f(s = \{x_1, x_2\}, x_i = \{0, 0\}) = (x_1 - 0)^2(x_2 - 0)^2$, $x_i \in [-1, 1]$. $L$ is set to 2. We first transfer the initial population with four individuals to the tensor and then sort and pad it into the new tensor with 16 channels. The modified DSC operator is employed to generate $x_1$, $x_2$, and the fitness tensor. $x_1$ and $x_2$ are handled by parameters shared across channels within a $3 \times 3$ convolution kernel. The fitness tensor is handled by Eq. (3). The new tensors of $x_1$, $x_2$, and the fitness tensor are averaged to generate the output.

### 3.5 TRAINING OF DECN

DECN with $t$ EMs generates the offspring $S_t$ from the input population $S_0$ and can be trained based on end-to-end mode. Then, given a proper loss function and training dataset, DECN can be trained to learn optimization strategies towards the objective function $f(s|\xi)$ by the back-propagation. We will generally establish a function set $F^{train}$ to train DECN.

**Training Dataset**. This paper establishes the training set by constructing a set of differentiable functions related to the optimization objective. This training dataset only contains $(S_0, f_i(s|\xi))$, the initial population and objective function, respectively. $f_i$ represents the $i$th function in this set. We show the designed training and testing datasets as follows:

$$F^{train} = \{f_1(s|\xi^{train}_{1,i}), \cdots, f_m(s|\xi^{train}_{m,i})\}, F^{test} = \{F_1(s|\xi^{test}_1)\} \qquad (4)$$

where $F_1$ is not employed in the training stage, and $m$ is the number of functions in $F^{train}$. $\xi^{train}_{m,i}$ represents the $i$th different values of $\xi$ in $m$th function $f_m$, which is true for any index pair. The initial population $S_0$ is always randomly generated before optimization. $F^{train}$ is comprised of different functions and has diverse landscapes from $F^{test}$.

**How to Train DECN**. DECN attempts to search for individuals with high quality based on the available information. The loss function tells how to adaptively adjust the DECN parameters to generate individuals closer to the optimal solution. According to the Adam (Kingma & Ba, 2014) method, a minibatch $\Omega$ is sampled each epoch for the training of DECN upon $F^{train}$, which is comprised by employing $K$ initialized $S_0$ for each $f_i$. We give the corresponding mean loss of minibatch $\Omega$ for $f_i$ in $F^{train}$,

$$\min_\theta \mathcal{L}_i = \min_\theta -\frac{1}{K} \sum_{S_0 \in \Omega} \frac{\frac{1}{|S_0|} \sum_{s \in S_0} f_i(s|\xi) - \frac{1}{|G_\theta(S_0)|} \sum_{s \in G_\theta(S_0)} f_i(s|\xi)}{\left| \left( \frac{1}{|S_0|} \sum_{s \in S_0} f_i(s|\xi) \right| \right.} \qquad (5)$$

Eq. (5) is to maximize the difference between the initial population and the output population of DECN to ensure that the initial population is close to the optimal solution. Moreover, Eq. (5)

---

**Algorithm 1** Training of DECN

---

**Input:** Batch size for Adam, $\Omega$; Function set for training, $F^{train}$
**Output:** Parameters of DECN $\theta$;
Randomly initialize $\theta$ of DECN;
Randomly initialize $\xi_{j,i}^{train}$ to adjust $f_j$ in $F^{train}$;
**repeat**
    Randomly initialize a minibatch $\Omega$ comprised of $K$ populations $S_0$;
    **for** $f_j$ **in** $F^{train}$ **do**
        Update $\theta$ by $\mathcal{L}_i$ given training data $(S_0, f_j)$;
    **end for**
    Update $\theta$ by minimizing $-1/m \sum_j \mathcal{L}_j$;
    Re-initialize parameters $\xi_j$ of $f_j$ in $F^{train}$ every $T$ epochs;
**until** training is finished

---

are generally differentiable based on the constructed training dataset. Eq. (5) enables DECN to perform the exploitation operation well but does not strongly encourage DECN to explore the fitness landscape. However, we have many options to balance exploration and exploitation. For example, the constructed Bayesian posterior distribution over the global optimum (Cao & Shen, 2020) is added to Eq. (5). Suppose the objective functions are hard to be formulated, and the derivatives of these functions are not available at training time, then two strategies can be employed: 1) Approximate these derivatives via REINFORCE (Williams, 1992); 2) Use the neuro-evolution (Such et al., 2017; Stanley et al., 2019) method to train DECN. Algorithm 1 provides the training process of DECN. After DECN has been trained, DECN can be used to solve the black-box optimization problems since the gradient is unnecessary during the test process.

Table 1: The compared results on six functions. The value of the objective function is shown in the table, and the optimal solution is bolded. *(*) represents the mean and standard deviation of repeated experiments.

| $D$ | $F$ | DECNws3 | DE | ES | CMA-ES | L2O-swarm | Dragonfly |
|---|---|---|---|---|---|---|---|
| | F4 | **1e-6(1e-07)** | 0.15(0.06) | 0.14(0.07) | 4e-3(4e-3) | 0.30(0.01) | 1310(1310) |
| | F5 | **1e-3(6e-06)** | 4.62(1.07) | 0.47(0.13) | 0.11(0.05) | 0.25(8.70e-4) | 48.4(9.58) |
| | F6 | **8.89(1e2)** | 244(95.1) | 118(235) | 1660(2540) | 154(243) | 4e8(1e08) |
| 10 | F7 | **0(0)** | 18.5(3.51) | 48.1(8.68) | 48.3(7.47) | 12.8(5.49) | 81.1(24.0) |
| | F8 | **0(0)** | 0.26(0.12) | 0.30(0.17) | 0.02(0.01) | 0.06(3.39e-4) | 35.4(22.0) |
| | F9 | **1e-3(3e-4)** | 1.86(0.36) | 20.5(0.13) | 20.7(0.10) | 2.19(0.02) | 16.2(3.64) |
| | F4 | **7e-9(4e-11)** | 8.5e3(396) | 9.3e4(8.12e3) | 7330(1e3) | 0.66(0.04) | None |
| | F5 | **4e-4(0)** | 28.2(0.47) | 82.5(2.16) | 71.8(9.74) | 0.96(0.04) | None |
| | F6 | **96(0.02)** | 2.3e8(2.8e7) | 2.5e10(3.1e9) | 3.4e8(9.5e7) | 286(28.5) | None |
| 100 | F7 | **0(0)** | 9.3e3(504) | 9.3e4(9.1e3) | 8.7e3(1.5e3) | 50.6(21.0) | None |
| | F8 | **0(0)** | 3.06(0.22) | 24.2(2.91) | 2.90(0.31) | 0.15(1e-3) | None |
| | F9 | **4e-3(8e-05)** | 18.9(0.15) | 21.4(0.01) | 21.4(0.03) | 3.06(0.02) | None |

## 4 EXPERIMENTS

### 4.1 RESULTS ON SYNTHETIC FUNCTIONS

**Results on High-fidelity Training Dataset** For each function in Appendix Table 7, we produce the training dataset as follows: 1) Randomly initialize the input population $S_0$; 2) Randomly produce a shifted objective function $f_i(s|\xi)$ by adjusting the corresponding location of optima-namely, adjusting the parameter $\xi$; 3) Evaluate $S_0$ by $f_i(s|\xi)$; 4) Repeat Steps 1)-3) to generate the corresponding dataset. For example, we show the designed training and testing datasets for the F4 function as follows:

$$F^{train} = \{F4(s|\xi_1^{train}), \cdots, F4(s|\xi_m^{train})\}, \quad F^{test} = \{F4(s|\xi^{test})\} \quad (6)$$

$F^{train}$ and $F^{test}$ are comprised of the same essential function but vary in the location of optima obtained by setting different combinations of $\xi$ (called $b_i$ in Table 7). $F^{train}$ can be considered

as the high-fidelity surrogate functions of $F^{test}$. We train DECN on $F^{train}$, and then we test the performance of DECN upon $F^{test}$, where the values of $\xi^{test}$ not appearing in the training process.

Here, $D = \{10, 100\}$ and $L = 10$. DECN is compared with standard EA baselines (DE (DE/rand/1/bin) (Das & Suganthan, 2010), ES (($\mu,\lambda$)-ES), and CMA-ES), L2O-swarm (Cao et al., 2019) (a representative L2O method for black-box optimization), and Dragonfly (Kandasamy et al., 2020) (the state-of-the-art Bayesian optimization). DECNws3 contains 3 EMs, and the parameters of these three convolution kernels are consistent across different EMs (weight sharing). The detailed parameters of these models can be found in Appendix A.11. The results are provided in Table 1. DECN outperforms compared methods by a large margin. This is because we use a high-fidelity surrogate function of the target black-box function to train DECN. The trained DECN contains an optimization strategy that is more tailored to the task. Current DE, ES, CMA-ES, and Dragonfly do not use this information to design their element. Even if we constantly adjust the hyperparameters of the comparison algorithm, the results are unlikely to be better than DECN.

**Results on Low-fidelity Training Dataset**
The training of DECN requires a differentiable surrogate function for the black-box optimization problem. However, accurate high-fidelity surrogate functions are difficult to obtain. Therefore, this section tests the performance of DECNs trained on low-fidelity surrogate functions. Three functions in Table 6 are employed as the low-fidelity surrogate functions for each function in Table 7. Here, the whole functions in Table 6 are employed as $F^{train}$ in order to train one DECN, and then the results on each function of Table 7 are shown in Table 2. For example, we show the designed training and testing datasets for the F4 function as follows:

Table 2: The performance of different DECN.

| $D$ | F | DECNws3 | DECNws30 | DECNn15 |
|---|---|---|---|---|
| 10 | F4 | 53.8(14.3) | 1.17(0.58) | **0.09(0.02)** |
| | F5 | 4.26(0.61) | 0.59(0.16) | **0.19(0.03)** |
| | F6 | **4.5(2.3e4)** | 131(67.1) | 17.0(2.88) |
| | F7 | 24.4(4.01) | **0.35(0.22)** | 5.93(1.40) |
| | F8 | 1.48(0.14) | 0.29(0.08) | **0.17(0.04)** |
| | F9 | 4.33(0.41) | 0.91(0.32) | **0.22(0.04)** |
| 100 | F4 | 1.15e4(744) | 2.19e3(148) | **67.4(9.09)** |
| | F5 | 25.0(0.92) | 10.4(0.39) | **2.22(0.15)** |
| | F6 | 3e8(4e7) | 1e7(2e6) | **2e4(5e3)** |
| | F7 | 776(19.2) | 549(18.3) | **81.5(11.9)** |
| | F8 | 105(6.12) | 20.9(1.17) | **1.58(0.08)** |
| | F9 | 11.6(0.21) | 6.79(0.14) | **3.77(0.16)** |

$$F^{train} = \{F1(s|\xi_{1,i}), F2(s|\xi_{2,i}), F3(s|\xi_{3,i})\}, \quad F^{test} = \{F4(s|\xi^{test})\} \tag{7}$$

Meanwhile, we also test the impact of different architectures on DECN, including the different number of layers and whether weights are shared between layers. We design three models, including DECNws3, DECNn15, and DECNws30. DECNn15 does not share parameters across 15 EMs. DECNws30 shares parameters across 30EMs. Their parameters are shown in Table 9 (Appendix).

DECNws30 outperforms DECNws3 in all cases, demonstrating that deep architectures have stronger representation capabilities and can build more accurate mapping relationships between random populations and optimal solutions. DECNn15 outperforms DECNws3 and DECNws30 when $D = 100$. This case is more complex than the case with $D = 10$. Although the number of layers of DECNn15 is lower than that of DECNws30, its representation ability is stronger than that of DECNws30 because it does not share weights. However, when the number of layers becomes larger, this architecture is more difficult to train. The transferability of DECN is proportional to the fitness landscape similarity between the training set and the problem. When new problem attributes are not available in the training set, DECN can still perform better. However, if extreme attributes are not available, then DECN can be the less satisfactory performance for functions with this attribute. These results show that the optimization strategy learned by DECN has good generality and is transferable to many unseen objective functions.

## 4.2 RESULTS ON PROTEIN DOCKING

Protein docking predicts the 3D structures of protein-protein complexes given individual proteins' 3D structures or 1D sequences (Smith & Sternberg, 2002). We consider the *ab initio* protein docking problem, which is formulated as optimizing the Gibbs binding free energy for conformation $s$: $f(s) = \nabla G(s)$. We calculate the energy function in a CHARMM 19 force field as in (Moal & Bates, 2010). We parameterize the search space as $s \in \mathbb{R}^{12}$ as in (Cao & Shen, 2020). We only consider 100 interface atoms. The training set includes 125 instances (see Appendix A.9), which contains 25 protein-protein complexes from the protein docking benchmark set 4.0 (Hwang et al., 2010), each

of which has five starting points (top-5 models from ZDOCK (Pierce et al., 2014)). The testing set includes three complexes (with one starting model each) of different levels of docking difficulty. 1ATN is the protein class that appeared during training. 1ATN_7 is the No. 7 instance of the 1ATN class, and it did not appear in the training process. 2JEL_1 and 7CEI_1 are the No. 1 instances of the two classes of proteins that did not participate in the training process. For example, we show the designed training and testing datasets for 1ATN_7 as follows:

$$F^{train} = \{f(s|\xi_1), \cdots, f(s|\xi_{125})\}, \ F^{test} = \{f(s|\xi^{test})\} \tag{8}$$

where $\xi$ represents different instances of protein-protein complexes.

In L2O-swarm, the part of Bayesian posterior distribution over the global optimum (Cao & Shen, 2020) is removed to keep fair with DECN. The experimental results reported in Table 3 demonstrate that DECN outperforms L2O-swarm, Dragonfly, DE, CMA-ES, and ES in all three cases. Since L2O-swarm has no elite retention mechanism, its result is worse than the optimal value of the initial population.

Table 3: The compared results on *ab initio* protein docking problem. $D = 12$.

| Methods | 1ATN_7 | 2JEL_1 | 7CEI_1 |
|---|---|---|---|
| L2O-Swarm | 2091(25.08) | 2766(24.80) | 1690(23.64) |
| CMA-ES | -6240(100) | -6260(51.8) | -6170(18.4) |
| ES | -6200(48.1) | -6210(5.05) | -6180(2.47) |
| DE | -6260(58.1) | -6220(29.2) | -6140(20.5) |
| Dragonfly | -6160(4.3) | -6120(2.9) | -6103(2.0) |
| DECNws3 | **-6261(96.71)** | **-6250(84.38)** | **-6193(84.66)** |

### 4.3 RESULTS ON PLANAR MECHANICAL ARM

The planner mechanic arm has been frequently employed as an optimization problem to assess how well the black-box optimization algorithms perform (Cully et al., 2015; Vassiliades et al., 2018; Vassiliades & Mouret, 2018; Mouret & Maguire, 2020). The planner mechanic arm problem has two key parameters: the set of $L = (L_1, L_2, \cdots, L_n)$ and the set of angles $\alpha = (\alpha_1, \alpha_2, \cdots, \alpha_n)$, where $n$ represents the number of segments of the mechanic arm, and $L_i \in (0, 10)$ and $\alpha_i \in (-\Pi, \Pi)$ represent the length and angle of the $i$th mechanic arm, respectively. This problem is to find the suitable sets of $L$ and $\alpha$ such that the distance $f(L, \alpha, p)$ from the top of the mechanic arm to the target position $p$ is the smallest, where $f(L, \alpha, p) = \sqrt{\left(\sum_{i=1}^{n} \cos(\alpha_i)L_i - p_x\right)^2 + \left(\sum_{i=1}^{n} \sin(\alpha_i)L_i - p_y\right)^2}$, and $(p_x, p_y)$ represents the target point's x- and y-coordinates. Here, $n = 100$. We design two groups of experiments.

Table 4: The results of planar mechanical arm. $gen$ is the number of generations for EAs.

| Case | $gen$ | $r$ | DE | ES | CMA-ES | L2O-Swarm | DECNws3 |
|---|---|---|---|---|---|---|---|
| SC | 10 | 100 | 2.96(1.63) | 11.2(4.70) | 236(46.8) | 40.4(3.89) | **0.42(0.22)** |
| | | 300 | 11.3(14.7) | 45.3(43.3) | 243(125) | 69.5(3.77) | **1.04(1.25)** |
| | 50 | 100 | 1.28(0.60) | 10.7(5.91) | 2.42(0.65) | 40.4(3.89) | **0.42(0.22)** |
| | | 300 | 1.54(0.89) | 42.0(41.0) | 4.06(6.54) | 69.5(3.77) | **1.04(1.25)** |
| | 100 | 100 | 1.20(0.64) | 10.6(5.58) | 1.36(0.35) | 40.4(3.89) | **0.42(0.22)** |
| | | 300 | 1.38(0.71) | 44.9(43.3) | 1.38(0.41) | 69.5(3.77) | **1.04(1.25)** |
| CC | 100 | 100 | 0.81(0.47) | 8.95(6.42) | 0.76(0.20) | 31.9(1.78) | **0.38(0.25)** |
| | | 300 | 6.15(12.2) | 47.8(56.0) | **0.87(0.37)** | 89.1(1.96) | 8.27(21.3) |

*1) Simple Case (SC).* We fixed the length of each mechanic arm as ten and only searched for the optimal $\alpha$. We randomly selected 600 target points within the range of $r \leq 1000$, where $r$ represents the distance from the target point to the origin of the mechanic arm, as shown in Fig. 10 (Appendix). In the testing process, we extracted 128 target points in the range of $r \leq 100$ and $r \leq 300$, respectively, for testing. We show the designed training and testing datasets as follows:

$$F^{train} = \{f(\alpha|\xi_1), \cdots, f(\alpha|\xi_{600})\}, \ F^{test} = \{f(\alpha|\xi_1^{test}), \cdots, f(\alpha|\xi_{128}^{test})\}, \ \xi = (p_x, p_y) \tag{9}$$

*2) Complex Case (CC).* We need to search for $L$ and $\alpha$ at the same time. We show the designed training and testing datasets as follows:

$$F^{train} = \{f((L,\alpha)|\xi_1), \cdots, f((L,\alpha)|\xi_{600})\}, F^{test} = \{f((L,\alpha)|\xi_1^{test}), \cdots, f((L,\alpha)|\xi_{128}^{test})\} \tag{10}$$

We evaluate the performance of the algorithm by $\sum_{f \in F^{test}} f/128$. The experimental results are shown in Table 4. Note that Dragonfly performs poorly due to the high dimensional of this problem ($D = \{100, 200\}$. In simple cases, DECNws3 outperforms all baselines. Nevertheless, for complex cases, DECNws3 outperforms all baselines when $r \le 100$. However, when $r \le 300$, DECNws3 outperforms ES and L2O-Swarm and is weaker than DE and CMA-ES. as shown in Table 2, the performance of DECNws3 is worse than DECNws30 and DECNn15. When we use DECNn15 to optimize the complex case, its result is 0.54(0.26), which is better than all baselines.

## 4.4 ACCELERATING DECN WITH GPU

We show the surprising performance of DECN with GPU-accelerated CRM and SM. To display the adaptability of DECN to GPUs, we offer the average runtime (second) of DECN and unaccelerated EA for three generations in Table 5. See Appendix A.6 for more results. DECN and EA optimize $K = 32$ populations, each containing $L \times L$ individuals (number of individuals: $K \times L \times L$). Similarly, we employ the runtime of EA with SBX crossover and Breeder mutation operator without acceleration as a reference in this experiment. In the unified test environment, the function estimation time consumed by DECN and EA is basically the same. As can be seen, with the increase of $L$, the ad-

Table 5: DECN's calculation efficiency upon one 1080Ti GPU.

| $D$ | Algorithm | $L$ | |
|---|---|---|---|
| | | 10 | 80 |
| 10 | DECN | 0.0054 | 0.0500 |
| | EA | 0.6942 | 72.3664 |
| 50 | DECN | 0.008026 | 0.237182 |
| | EA | 0.6820 | 71.4425 |
| 500 | DECN | 0.0412 | 2.7274 |
| | EA | 0.7208 | 74.9417 |

vantage of acceleration based on GPUs is clear. DECN is around 102 times faster than EA when $D \in \{10, 50, 500\}$. This case indicates that DECN is adapted to the acceleration of GPUs and can be accelerated sufficiently. However, GPU cannot accelerate EA's crossover, mutation, and selection modules. In the case of a large population of individuals, these operators take up a high running time.

## 4.5 VISUALIZATION

We take a two-dimensional F4 function as an example to verify that DECN can indeed advance the optimization. In Fig. 4, as the iteration proceeds, DECN gradually converges. When passing through the first EM module, the CRM is first passed, and the offspring $S'_{i-1}$ are widely distributed in the search space, and the offspring are closer to the optimal solution. Therefore, the CRM generates more potential offspring and is rich in diversity. After the SM update, the generated $S_i$ is around the optimal solution, showing that the SM update can keep good solutions and remove poor ones. From the population distribution results of the 2nd, 3rd, and 15th EMs, DECN continuously moves the population to the vicinity of the optimal solution.

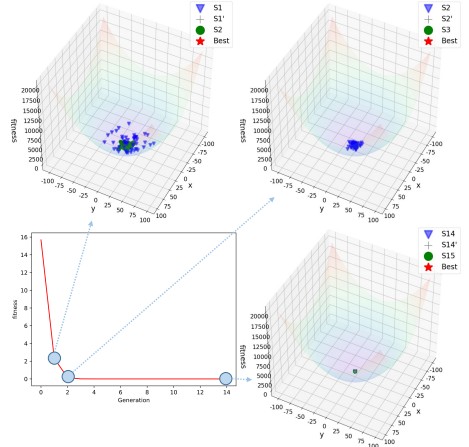

Figure 4: Visualization of the optimization process.

## 5 CONCLUSIONS

We successfully designed DECN to learn optimization strategies for black-box optimization automatically. The better performance than other human-designed methods demonstrates the effectiveness of DECN. DECN can be well adapted to new black-box optimization tasks. Moreover, DECN has outstanding adaptability to GPU acceleration due to the tensor operator. The limitations are discussed in Appendix A.8.

## 6 REPRODUCIBILITY STATEMENT

The source code of Pytorch version of DECN can be downloaded in supplementary materials. The parameters of DECN are shown in Table 9 in Appendix. Nine synthetic functions are shown in Appendix A.5 (Tables 6 and 7). The 25 protein-protein complexes used for training DECN are shown in Appendix A.9.

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

# A  APPENDIX

## A.1  BACKGROUND

**Recombination**. The subtraction is applicable during the production of a new individual, such as DE (Das & Suganthan, 2010) and recombination operator in EAs, usually conducted on $n$ individuals. DE can reproduce a unique individual $s^*$ based on 11 with $s_1, s_2, \cdots, s_n$.

$$s^* = s_k + \sum_{i=2}^{n-1} F_i(s_i - s_{i+1}) \tag{11}$$

where $F_i$ is a scaling factor and $s_k$ is the best solution or is selected from $s_1, s_2, \cdots, s_n$. After an expression expansion, this process can be summarized by a weighted recombination process as given in Eq. (12). These operators are manually designed with different parameters ($a_i$).

$$s^* = a_1 \times s_1 + a_2 \times s_2 + \cdots + a_n \times s_n = \sum_{i=1}^{n} a_i \times s_i. \tag{12}$$

**Selection**. Many selection operators exist, such as the binary tournament mating selection operator in Eq. (13). The selection operator is to retain individuals of higher quality for the next generation, which can be regarded as an information selection process.

$$p_i = \begin{cases} 1 & f(s_i) < f(s_k) \\ 0 & f(s_i) > f(s_k) \end{cases}, \quad (s_i, s_k) \in S, \tag{13}$$

where $p_i$ reflects the probability that $s_i$ is selected for the next generation, and $(s_i, s_k)$ in Eq. (13) are randomly selected from the population $S$. The selection process will be repeated until the number of individuals is chosen.

## A.2  HOW TO ORGANIZE A POPULATION INTO A TENSOR

As shown in Fig. 5, individuals in the lattice are sorted in descending order to construct a population tensor with a consistent pattern. Suppose a population $S = s_1, s_2, \cdots, s_{L \times L}$ and $f(s_1) < f(s_2) < \cdots < f(s_{L \times L})$, where $f(s)$ is a minimization task. $s_1, s_2, \cdots, s_{L \times L}$ are arranged in descending order within the $L \times L$ lattice.

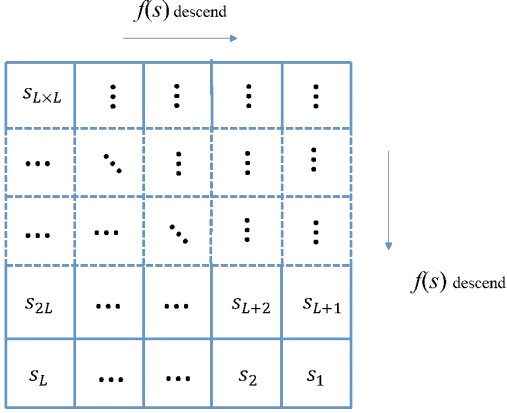

Figure 5: Organizing a population into a tensor.

Figure 6 gives an illustration of the tensor data. As can be seen, the number of channels of input tensors is $D+1$, where $D$ is the dimension of the optimization task, and the fitness of individuals occupies one channel. The fitness channel does not participate in the convolution process but is essential for the selection module in DECN for the information selection.

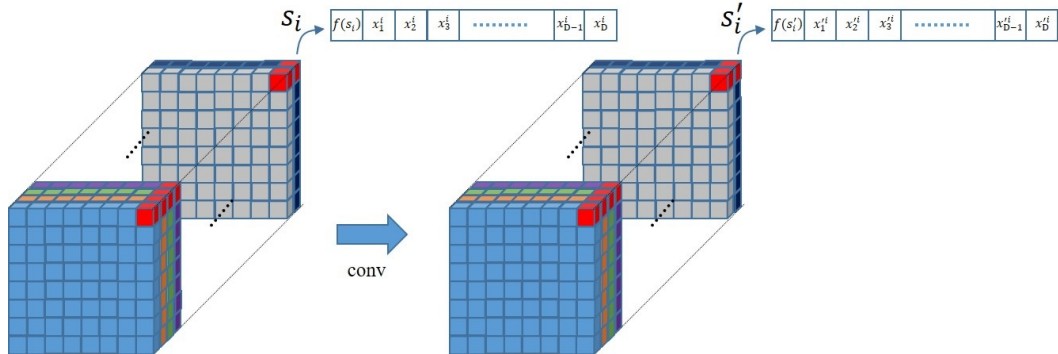

Figure 6: The realization of similar functions as recombination operators based on convolution operator. Convolution kernels slip over the whole $L \times L$ lattice and conduct the information interaction within the neighborhood of $(i, j)$. For the picture on the left, the small red square with many channals represents $S_i$.

## A.3 POPULATION ARRANGEMENT

Figure 7 gives an example of population arrangement and padding for the problem $\min f(s) = x_1 \times x_1$, where $s = x_1, x_1 \in [0, 10]$. The blue part marks the population arranged in a $10 \times 10$ lattice, while the gray region marks the mirror padding part.

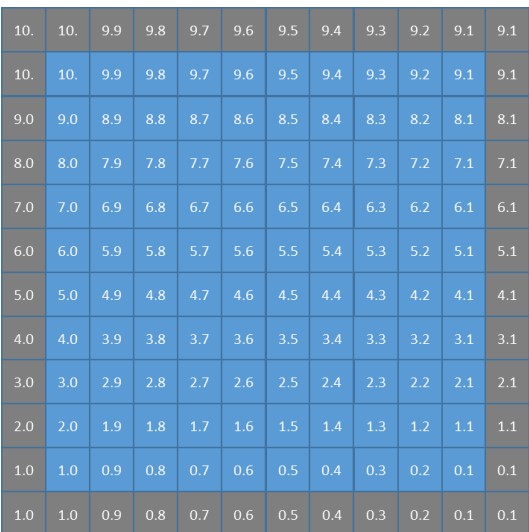

Figure 7: Population arrangement and padding.

## A.4 SEVERAL ESSENTIAL ISSUES ABOUT CRM

There are several essential issues necessary to be considered.

1) **How many individuals should participate in the CRM reasoning progress**. It remains a challenge to implement information reasoning over multi-individuals in EAs. In most recombination operators, the participant number is usually set to 2. However, based on the gradient information provided by the back-propagation, it is easy to control an individual's element by adjusting $w_{k,l}$.

2) **How to integrate the offspring produced by different convolution kernels**. Since the convolution operation can be transformed as a multiplication between matrices, simply averaging over the

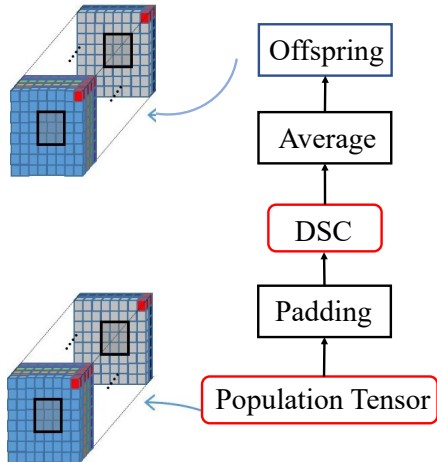

Figure 8: Reproduction of the offspring based on CRM.

results output by different convolution kernels does not influence the training process. For example, $a_1Co^1x_1^i + a_2Co^2x_2^i + a_3Co^3x_3^i \leftrightarrow Co^{'1}x_1^i + Co^{'2}x_2^i + Co^{'3}x_3^i$, where $x_1^i$, $x_2^i$, and $x_3^i$ are input elements of $s_i$, $a_1$, $a_2$, and $a_3$ are the constant, and $Co$ denotes the convolution matrix.

3) **How many convolution kernels should be used within CRM**. We suppose that these are three convolution kernels for $x$. We can find that the outcome $a_1Co_{3\times3}^1x + a_2Co_{3\times3}^2x + a_3Co_{3\times3}^3x$ is equivalent to $a^{'}Co_{3\times3}^{'}x$. The output of multiple convolution kernels can be replaced by one convolution kernel. Thus, the number of convolution kernels of the same size has no apparent influence on DECN.

4) **The impact of neighborhood recombination operation**. The neighborhood recombination operation has been commonly accepted in EAs to alleviate the selection pressure and prevent the premature convergence of populations. Moreover, the receptive field of convolution kernels expands as the number of layers increases. Thus, DECN can learn efficient optimization strategies across generations.

## A.5 NINE SYNTHETIC FUNCTIONS AND PARAMETERS

Table 6: Training functions.

| ID | Functions | Range |
|----|-----------|-------|
| F1 | $\sum_i |w_i sin(x_i - b_i)|$ | $x \in [-10, 10], b \in [-10, 10]$ |
| F2 | $\sum_i |x_i - b_i|$ | $x \in [-10, 10], b \in [-10, 10]$ |
| F3 | $\sum_i |(x_i - b_i) - (x_{i+1} - b_{i+1})| + \sum_i |x_i - b_i|$ | $x \in [-10, 10], b \in [-10, 10]$ |

## A.6 ACCELERATE DECN WITH GPU

The acceleration of EAs using GPUs is challenging, and lots of research has contributed to this problem. The support for multiple subpopulations to evolve simultaneously has paramount significance in practical applications. The efforts (Jin & Qin, 2017; Qin et al., 2012) accelerated the K-Means process within the brain storm optimization algorithm through GPUs and proposed an improved CUDA-based implementation of differential evolution on GPUs. Many other EAs have benefited from the computing performance of GPUs (Huang et al., 2021; Cheng & Gen, 2019). However, all of them just parallelized the current EAs. Besides, many available genetic operators are unfriendly to the GPU acceleration, as GPUs are weak in processing logical operations. As both CRM and SM are comprised of operations upon tensors, they can be sufficiently accelerated by GPUs.

Table 7: Testing Functions.

| ID | Functions | Range |
|---|---|---|
| F4(Sphere) | $\sum_i z_i^2, z_i = x_i - b_i$ | $x \in [-100, 100], b \in [-50, 50]$ |
| F5 | $\max\{|z_i|, 1 \le i \le D\}, z_i = x_i - b_i$ | $x \in [-100, 100], b \in [-50, 50]$ |
| F6(Rosenbrock) | $\sum_{i=1}^{D-1} (100(z_i^2 - z_{i+1})^2 + (z_i - 1)^2), z_i = x_i - b_i$ | $x \in [-100, 100], b \in [-50, 50]$ |
| F7(Rastrigin) | $\sum_{i=1}^{D} (z_i^2 - 10\cos(2\pi z_i) + 10), z_i = x_i - b_i$ | $x \in [-5, 5], b \in [-2.5, 2.5]$ |
| F8(Griewank) | $\sum_{i=1}^{D} \frac{z_i^2}{4000} - \prod_{i=1}^{D} \cos(\frac{z_i}{\sqrt{i}}) + 1, z_i = x_i - b_i$ | $x \in [-600, 600], b \in [-300, 300]$ |
| F9(Ackley) | $-20\exp(-0.2\sqrt{\frac{1}{D}\sum_{i=1}^{D} z_i^2})$ $-$ $\exp(\frac{1}{D}\sum_{i=1}^{D}\cos(2\pi z_i)) + 20 + \exp(1), z_i = x_i - b_i$ | $x \in [-32, 32], b \in [-16, 16]$ |

Table 8: Investigation of DECN's calculation efficiency when accelerated upon one 1080Ti GPU. The results in this table are the average time (second) of algorithms to conduct the evolution of 32 input populations for three generations.

| D | Algorithm | L | | | |
|---|---|---|---|---|---|
| | | 10 | 20 | 40 | 80 |
| | DECN(s) | 0.004627 | 0.005978 | 0.007449 | 0.015492 |
| 2 | EA(s) | 0.700342 | 2.863495 | 12.67563 | 71.74005 |
| | Rate(DECN/EA) | 0.006607 | 0.002088 | 0.000588 | 0.000216 |
| | DECN | 0.005487 | 0.007838 | 0.01664 | 0.049973 |
| 10 | EA | 0.694213 | 2.879791 | 12.87636 | 72.3664 |
| | Rate(EM/EA) | 0.007904 | 0.002722 | 0.001292 | 0.000691 |
| | DECN | 0.007052 | 0.013323 | 0.039877 | 0.138544 |
| 30 | EA | 0.693307 | 2.87876 | 13.00381 | 71.67544 |
| | Rate(EM/ EA) | 0.010171 | 0.004628 | 0.003067 | 0.001933 |
| | DECN | 0.008026 | 0.01875 | 0.062079 | 0.237182 |
| 50 | EA | 0.681967 | 2.868 | 12.80662 | 71.44253 |
| | Rate(EM/ EA) | 0.011769 | 0.006538 | 0.004847 | 0.00332 |
| | DECN | 0.011725 | 0.033109 | 0.117593 | 0.478518 |
| 100 | EA | 0.699074 | 2.865546 | 13.12218 | 71.83043 |
| | Rate(EM/ EA) | 0.016772 | 0.011554 | 0.008961 | 0.006662 |
| | DECN | 0.041167 | 0.147843 | 0.610056 | 2.727426 |
| 500 | EA | 0.720847 | 2.966926 | 13.59977 | 74.9417 |
| | Rate(EM/ EA) | 0.057109 | 0.04983 | 0.044858 | 0.036394 |

DECN mainly containing operations upon tensors and is easily accelerated by GPUs. Current distributed EA methods usually separate a population into multiple subpopulations that evolve simultaneously. Such separation is also a commonly accepted operation in many EAs However, none of them can accelerate the genetic operators. Here, we show the surprising performance of DECN with GPU accelerated CRM and SM. Moreover, Tensorflow has provided mature solutions for the acceleration upon GPUs, and DECN implemented by Tensorflow is supportable to load multiple populations as the input.

To show the adaptability of DECN to GPUs, we offer the runtime of DECN and unaccelerated EA in Table 8, within which both DECN and EA optimize $K = 32$ populations with each containing $L \times L$ individuals (number of individuals: $K \times L \times L$). Similarly, we employ the runtime of EASBX without acceleration as a reference in this experiment. As can be seen, with the increase of $L$, the advantage of acceleration based on GPUs is clear. When the dimension $D \in \{2, 10\}$, DECN runs 103~104 times faster than EA. DECN is still around 102 times faster than EA when $D \in \{30, 50, 100, 500\}$. This case indicates that DECN is adapted to the acceleration of GPUs and can be accelerated sufficiently. However, with increasing $D$, DECN increases the proportion of evaluations in the runtime and ultimately weakens the advantage of acceleration. These cases indicate the acceleration advantage of DECN when optimizing a larger population.

### A.7 The Convergence of Loss Function in Training Process

This part is the change curve of the loss function of the training process of DECNnws15 on F4-F9. The results are shown in Fig. 9. Here, $D$=10.

### A.8 Limitations

However, DECN has many drawbacks. We hope to address these deficiencies in future work. 1) The designed loss function enables DECN to perform the exploitation operation well but does not strongly encourage DECN to explore the fitness landscape. However, we have many options to balance exploration and exploitation. For example, the constructed Bayesian posterior distribution (Cao & Shen, 2020) over the global optimum is added to Eq. 7. In addition to adding items that focus on the exploration ability of the loss function, new modules can also be designed to be added to the EM to help DECN jump out of the local optimum.

2) For the constructed training dataset, DECN does not have an advantage if it is utterly irrelevant to the optimization objective. Thus, establishing a suitable training dataset is essential.

3) DECN only focuses on continuous optimization problems without constraints. For problems such as expensive optimization, combinatorial optimization, constrained optimization, and multi-objective optimization, DECN needs to be adjusted according to the characteristics of the problem. We can think of DECN as standard optimizers like vanilla DE, ES, GA, and PSO. In order to deal with different types of problems, we need to make different corrections to DECN. For example, to deal with expensive problems, we need to build surrogate models to assist DECN. We need to redesign the CRM module to generate new feasible solutions for combinatorial optimization problems. For example, for TSP tasks, GNN may be a feasible option to generate new solutions instead of CRM. We can redesign the CRM module for constrained optimization problems to generate feasible solutions. Of course, the easiest way is to use constraint violations and fitness functions as criteria for selecting the next generation in the SM module.

### A.9 Training Dataset for Protein Docking

The training dataset contains 25 protein-protein complexes from the protein docking benchmark set 4.0 (Hwang et al., 2010). The detailed information is shown as follows: 1ATN, 1AVX, 1AY7, 1BJ1, 1BVN, 1CGI, 1DFJ, 1EAW, 1EWY, 1EZU, 1GRN, 1IBR, 1IJK, 1IQD, 1JPS, 1KXQ, 1M10, 1MAH, 1N8O, 1PPE, 1R0R, 1XQS, 2B42, 2C0L, and 2HRK.

### A.10 Motivations

The generalization ability of current evolutionary algorithms (EAs) is poor. Faced with a new black-box optimization task, we need experts to redesign/select the EA's crossover, mutation, and selection

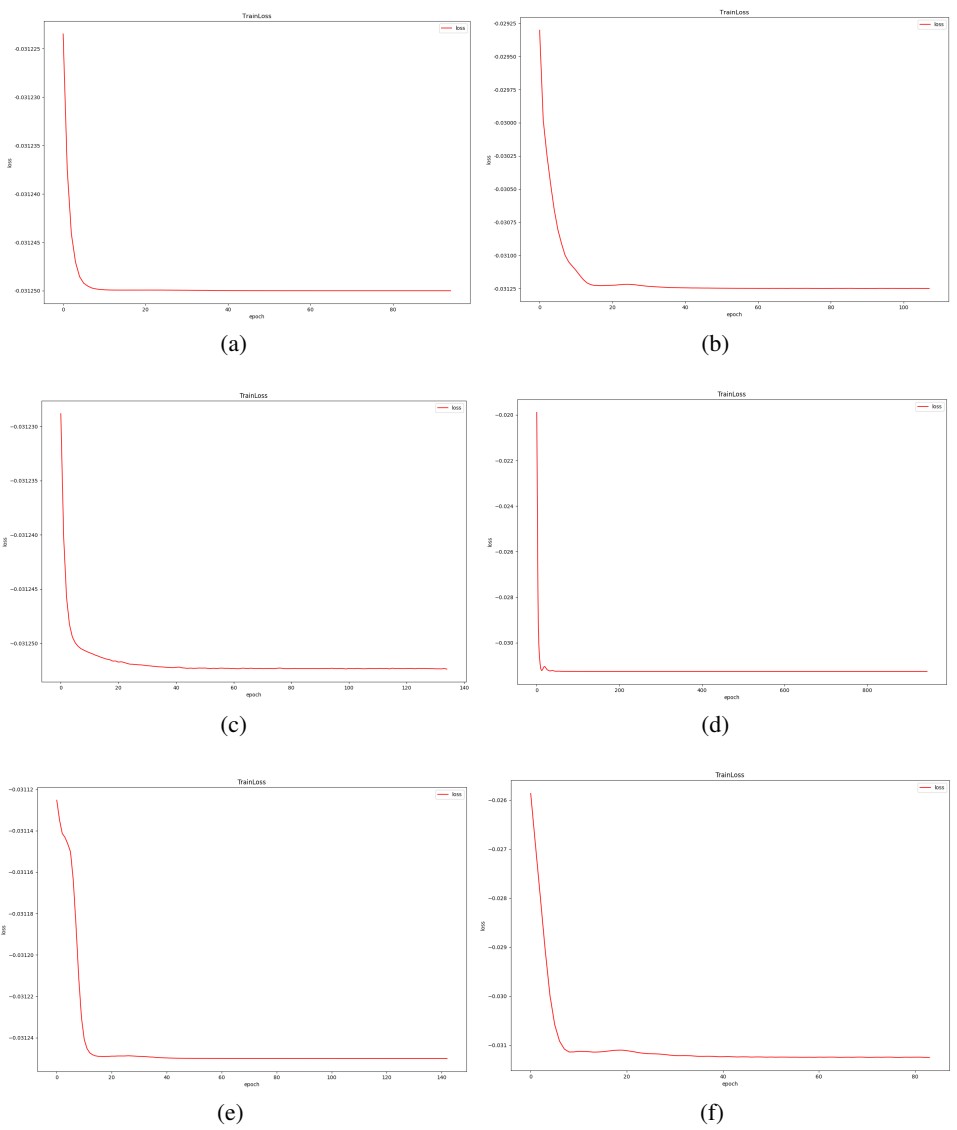

Figure 9: The convergence of loss function in training process. (a) F4, (b) F5, (c) F6, (d) F7, (e) F8, and (f) F9.

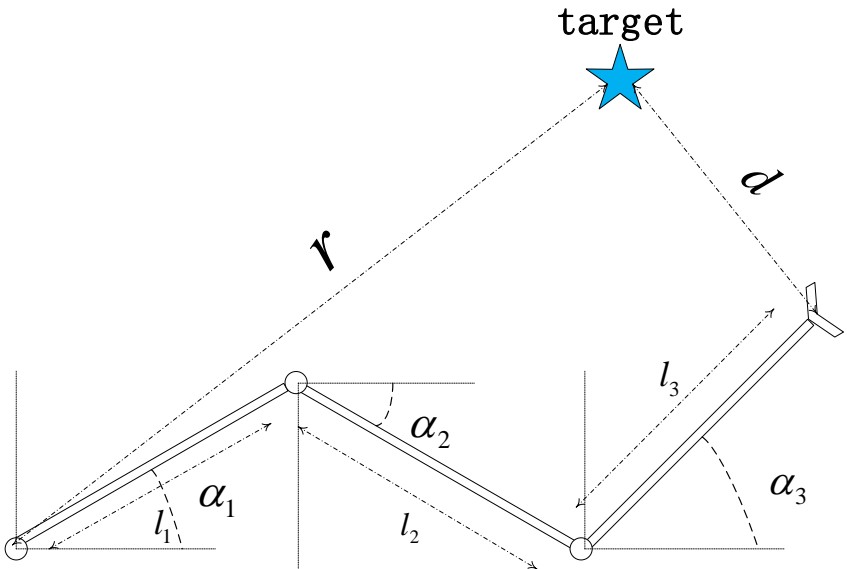

Figure 10: Planar Mechanical Arm.

operations (including their hyperparameters) to maximize its performance on the target task, resulting in a hand-designed EA with big application limitation. Most importantly, due to the limitation of expert knowledge, only little target function information is used to assist the design of EA, which makes it challenging to adapt to the target task. How to automatically design optimization strategies according to new tasks is crucial. To the best of our knowledge, there is currently no work to address this issue. We think EA is a generative optimization model that realizes the generation from a random population to an optimal solution by manually designing crossover, mutation, and selection operations. The purpose of these operations is to generate potential solutions and retain good solutions. The task of automatically designing an optimization strategy is learning how to automatically generate and retain potential solutions. This paper is to show how DECN finish this task.

By constructing a set of differentiable surrogate functions of the objective black-box function, DECN can allow the designed CRM and SM to learn the strategy of optimizing the objective function. At this point, DECN effectively utilizes the information of the target black-box function to assist the construction of the optimization strategy. The degree of fit of DECN with the target task is much higher than that of the human-designed EA. The following statement may be one-sided: Bayesian optimization also suffers from poor generalization. For example, how to choose/design appropriate acquisition functions for different problems.

We use the process of an evolutionary algorithm to guide the design of DECN and realize the mapping from a random population to the optimal solution.

First, we need to design a module to ensure the exchange of information between individuals in the population to achieve the function of generating potential solutions (similar to the recombination operators in EA). We can achieve this function by modifying the convolution operation accordingly, which is our motivation for using convolution to design. The designed CRM module achieves this purpose (see Section 3.1).

Second, to survive good individuals for the next layer of DECN, we design the selection module (SM) based on a pairwise comparison between the offspring and input population regarding their fitness (see Section 3.3, Equation 3). We can clearly observe that Equation 3 can indeed keep good individuals.

Third, the untrained DECN does not handle the black-box optimization problem well because it needs information about the target black-box function. In order to better optimize the objective

task, we need to design a training set containing objective function information and a practical loss function to guide the parameter training of DECN (see Section 3.5). The characteristics of black-box functions make it difficult for us to obtain their gradient information to assist in the training of DECN. We construct a differentiable surrogate function set of the target black-box function to obtain the information of the target black-box function. The designed loss function is to maximize the difference between the initial population and the output population of DECN to ensure that the initial population is close to the optimal solution.

There are few learning-to-optimize architectures (Chen et al., 2022) currently dealing with black-box optimization problems, and their performance is weak. From the experimental results, DECN makes up for the performance disadvantage of the learning-to-optimize architecture in the black-box optimization problem. We also strongly believe that this paper makes an essential contribution to the learning-to-optimize community.

### A.11   PARAMETERS

DECN is compared with standard EA baselines (DE (DE/rand/1/bin) (Das & Suganthan, 2010), ES ($(\mu,\lambda)$-ES), and CMA-ES), L2O-swarm (Cao et al., 2019) (a representative L2O method for black-box optimization), and Dragonfly (Kandasamy et al., 2020) (the state-of-the-art Bayesian optimization). DE and ES are implemented based on Geatpy (et.al., 2020), and CMA-ES is implemented by Pymoo (Blank & Deb, 2020). The parameters of DE, ES, CMA-ES, and Dragonfly are adjusted to be optimal for each problem. L2O-swarm and DECN use the same training set and loss function. All algorithms are run ten times for each function. DECNws3 contains 3 EMs, and the parameters of these three convolution kernels are consistent across different EMs (weight sharing). The population sizes of DE, ES, CMA-ES, and DECN are 100. DE, ES, and CMA-ES run for 100 generations. For DECNws3, its architecture determines that DECN has only been iterated for three generations. DE, ES, and CMA-ES have 100/3 times as many function evaluations as DECN, which is highly unfair to DECN. Both Dragonfly and L2O-Swarm run to convergence.

Table 9: Experimental setup for DECNws30, DECNws3 and DECNnws15. In DECNws3, parameters of these three convolution kernels are consistent across different EMs (weight sharing). Moreover, during the training process, the 2-norm of gradients is clipped to be not larger than 10, and the learning rate ($lr = 0.01$) shrinks every 100 epochs. The shrinking rate is set to 0.9. The generation of these reference algorithms is set to 100, while DECNws3 only evolves the population with 3 EMs. 5000 epochs are conducted during the training process. All experimental studies are performed on a Linux PC with Intel Core i7-10700K CPU at 3.80GHz and 32GB RAM.

| Model | $L$ | $D$ | $K$ | EMs | Convolution kernels | $lr$ | Epochs | $T$ | Weight share | Gradient norm |
|---|---|---|---|---|---|---|---|---|---|---|
| DECNws30 | 10 | 10 | 32 | 30 | $3 \times 3: u = 0, \sigma = 0.5$ $5 \times 5: u = 0, \sigma = 0.5$ $7 \times 7: u = 0, \sigma = 0.5$ | 0.01 | 10000 | 10 | True | 10 |
| DECNws3 | 10 | 2 | 32 | 3 | $3 \times 3: u = 0, \sigma = 0.5$ $5 \times 5: u = 0, \sigma = 0.5$ $7 \times 7: u = 0, \sigma = 0.5$ | 0.0005 | 5000 | 10 | True | 10 |
| DECNn15 | 10 | 30 | 16 | 15 | $3 \times 3: u = 0, \sigma = 0.5$ $5 \times 5: u = 0, \sigma = 0.5$ $7 \times 7: u = 0, \sigma = 0.5$ | 0.0005 | 2000 | 10 | False | 10 |

