# OpenReview forum: "DECN: Evolution Inspired Deep Convolution Network for Black-box Optimization"
_ICLR.cc/2023/Conference — Submitted to ICLR 2023_

### Official Review · Reviewer_g29J · 2022-10-23

**Confidence:** 4
**Correctness:** 3
**Technical Novelty And Significance:** 3
**Empirical Novelty And Significance:** 3
**Recommendation:** 6

**Clarity, Quality, Novelty And Reproducibility:**

**Clarity**

The clarity in the current draft is often weak, making it difficult to follow important parts of the paper. Furthermore, many of the tables and figures (including Table 2, 3, 4) do not have enough information to understand them on their own forcing the reader to track relevant sections of text which are sometimes not co-located.

**Quality**

Overall, the presented experiments appear sound, but it is difficult to judge their overall quality based the lack of clarity in some sections of the paper.

**Novelty**

Overall, the core part of the methodology (creating the convolution operator and associated data structure for the solution space) appears novel and relevant.

**Reproducibility**

The detailed descriptions provided in the paper should make it reasonable to reproduce the presented results. While the authors do not mention release of code explicitly in the paper it appears to be included in the supplementary material. Moreover, the current draft does not include a reproducibility statement that could clarify reproducibility questions.

**Strength And Weaknesses:**

**Strengths**

* The paper proposes a novel method to introduce learnable convolution operations into an evolution inspired setting. The results shown in the paper indicate that the proposed method outperforms some common black-box optimizers on a given set of tasks in addition to providing some generalization benefits.
* The papers provides a cohesive data structure for applying the method and outlines its different component in much detail.

**Weaknesses**

* The paper studies only limited setting (six standard black box optimization functions) with limited experimental settings. Given that the authors claim scalability and speed of their method as an advantage, it would have been nice to see more challenging optimization settings with greater compute costs.
* The protein docking experiments is not very well described. How does this setting turn into continuous optimization and what is the search space? It would also be helpful to have a consistent set of baselines across all experiments.
* The setting of the transferability study is unclear. Is this zero-shot learning or few-shot learning? It's hard to asses how meaningful the results are without those details.
* The authors dedicate a large part of the paper to describing many details of their methods, which then ends up taking up the majority of space. I think the paper could be strengthened by focusing the method on the important parts and re-using the rest of the paper to perform a more thorough analysis.

**Additional Questions**

* Are you initializing multiple populations at the same time? In Algorithm 1, you say "initialize a minibatch comprised of K populations", which appears to imply that there are multiple populations solving the same problem?
* What is $\zeta$ in equation 7? I did not find a definition.
* It would be good to clarify what DE is in your baselines.
* Did the GPU implementation rely purely on deep learning frameworks? Your general description seems to indicate that.




**Summary Of The Paper:**

The authors propose a deep convolution network (DECN) that mimics the operation of evolutionary search techniques, specifically recombination and selection, for black-box optimization problems. The paper first describes the general challenges in creating a convolution operator to mimic evolutionary, as well as related work, and then provides definitions of the operations the authors aim to achieve in DECN design (recombination and selection). Subsequently, the authors describe their method, including how they construct a tensor from a population of solutions for a given black-box optimization problem and arrange it to make it amendable for learnable convolutions. The convolution reasoning module (CRM) serves to produce the offspring for the population, whose components are described in detail by the authors, while the selection module (SM), based on pairwise comparisons of solution fitnesses, serves to select the solutions of the evolving population. Taken together CRM and SM make up an evolution module (EM), which can then be used for end-to-end gradient based optimization of the DECN modules.

The authors then test DECN on six black-box optimization functions and compare DECN's performance to a set of black-box optimization algorithms with the Table 1 showing outperformance of DECN. Subsequently, authors show results on a protein docking task, provide a study of generalization of DECN module across the six aforementioned optimization functions, as well as GPU runtime study.

**Summary Of The Review:**

Overall, I would say that the weaknesses of the paper in its current form outweigh the strengths leading me to vote for rejection. My most pressing concerns include the lack of clarity in the way the paper is currently written (which affects many other parts of my assessment), as well as the quality and relevance of the current experiments (would like to see more challenging optimization problems and the inclusion of more modern baseline algorithms). I would say that the core novelty proposed in the method holds promise and would encourage the authors to continue to refine their research to make the paper stronger.

---
Updating my score during discussion phase given the changes and responses provided by the authors.

---

> ### Author Response · Authors · 2022-11-14
> **Response to Reviewer g29J (Part 2)**
>
> **Q4:** The authors dedicate a large part of the paper to describing many details of their methods, which then ends up taking up the majority of space. I think the paper could be strengthened by focusing the method on the important parts and re-using the rest of the paper to perform a more thorough analysis.
>
> **Answer 4:** Very good advice. We have made changes based on your suggestions, and the whole article is refreshing. At the same time, we put a more detailed discussion in Appendix. Please refer to Section 3 for the specific modification content. Thank you very much.
>
> **Q5:** Are you initializing multiple populations at the same time? In Algorithm 1, you say "initialize a minibatch comprised of K populations", which appears to imply that there are multiple populations solving the same problem?
>
> **Answer 5:** Yes, we take advantage of GPU parallelism to improve training speed.
>
> **Q6:** What is ζ in equation 7? I did not find a definition.
>
> **Answer 6:** $\xi$ is the known parameters of $f$. By perturbing $\xi$, we can get different fidelity differentiable surrogate models of the target black box function. For each experiment, we have given the specific training set and test set, as well as the corresponding definition of $\xi$.
> On synthetic black-box functions, $\xi$ is called $b_i$ in Tables \ref{table:test} and \ref{table:train}. In the protein docking problem, $\xi$ represents different protein-protein complexes. On the planner mechanic arm problem, $\xi$ is the target position $(p_x, p_y)$.
>
> **Q7:** It would be good to clarify what DE is in your baselines.
>
> **Answer 7:** DECN is compared with standard EA baselines (DE(DE/rand/1/bin), ES(($\mu$,$\lambda$)-ES), and CMA-ES). DE and ES are implemented based on Geatpy [1], and CMA-ES is implemented by Pymoo [2]. The parameters of DE, ES, CMA-ES, and Dragonfly are adjusted to be optimal for each problem.
>
> [1] Jazzbin et.al. geatpy: The genetic and evolutionary algorithm toolbox with high performance in python, 2020. http://www.geatpy.com/
>
> [2] J. Blank and K. Deb. pymoo: Multi-objective optimization in python. IEEE Access, 8:89497–89509, 2020
>
> **Q8:** Did the GPU implementation rely purely on deep learning frameworks? Your general description seems to indicate that.
>
> **Answer 8:** GPU implementations all use the deep learning framework they rely on. We also provide implementation code for Pytorch version of DECN (see Supplementary Materials).
>
> **Q9:** The clarity in the current draft is often weak, making it difficult to follow important parts of the paper. Furthermore, many of the tables and figures (including Table 2, 3, 4) do not have enough information to understand them on their own forcing the reader to track relevant sections of text which are sometimes not co-located.
>
> **Answer 9:** 1) We have revised Section 3 by removing redundant discussions and placing them in Appendix. The revised version allows readers to follow the DECN process clearly. Please move to Section 3 of the modified version to review whether our modification is appropriate.
>
> 2) We rewrite the experimental part. The main modifications are: (1) clearly give the purpose of each experiment; (2) clearly define the training set and test set of each question. Please move to Section 4 of the modified version to review whether our modification is appropriate.
>
> **Q10:** The detailed descriptions provided in the paper should make it reasonable to reproduce the presented results. While the authors do not mention the release of code explicitly in the paper, it appears to be included in the supplementary material. Moreover, the current draft does not include a reproducibility statement that could clarify reproducibility questions.
>
> **Answer 10:** The source code of Pytorch version of DECN can be downloaded in supplementary materials. The parameters of DECN are shown in Table 8 in Appendix. Nine synthetic functions are shown in Appendix A.5 (Tables 6 and 7). The 25 protein-protein complexes used for training DECN are shown in Appendix A.9.

---

> ### Author Response · Authors · 2022-11-14
> **Response to Reviewer g29J (Part 1)**
>
> **Q1:** The paper studies only limited settings (six standard black box optimization functions) with limited experimental settings. Given that the authors claim scalability and speed of their method as an advantage, it would have been nice to see more challenging optimization settings with greater compute costs.
>
> **Answer 1:** We test our proposal on a complex planar mechanical arm problem and consider two cases: 1) simple case (SC): the experiments of the planar mechanical arm on searching for different angles with the fixed lengths; 2) complex case (CC): the more challenging experiments of the planar mechanical arm on searching for different angles and lengths. The specific parameter settings are shown in the revised version of Section 4.3. The experimental results are shown as follows:
>
> |Case | $gen$ | $r$ | DE | ES| CMA-ES | L2O-Swarm | DECNws3|
> |- | - | -  | - |- | - | - | -|
> |SC | 10 | 100 | 2.96(1.63) | 11.2(4.70) | 236(46.8) | 40.4(3.89) | **0.42(0.22)**|
> |SC | 10 | 300 | 11.3(14.7) | 45.3(43.3) | 243(125) | 69.5(3.77) | **1.04(1.25)**|
> |SC | 50 | 100 | 1.28(0.60) | 10.7(5.91) | 2.42(0.65) | 40.4(3.89) | **0.42(0.22)**|
> |SC | 50 | 300 | 1.54(0.89) | 42.0(41.0) | 4.06(6.54) | 69.5(3.77) | **1.04(1.25)**|
> |SC | 100 | 100 | 1.20(0.64) | 10.6(5.58) | 1.36(0.35) | 40.4(3.89) | **0.42(0.22)**|
> |SC | 100 | 300 | 1.38(0.71) | 44.9(43.3) | 1.38(0.41) | 69.5(3.77) | **1.04(1.25)**|
> |CC | 100 | 100| 0.81(0.47) | 8.95(6.42) | 0.76(0.20) | 31.9(1.78) | **0.38(0.25)**|
> |CC | 100 | 300 | 6.15(12.2) | 47.8(56.0) | **0.87(0.37)** | 89.1(1.96) | 8.27(21.3) |
>
> In simple cases, DECNws3 outperforms all baselines. Nevertheless, for complex cases, DECNws3 outperforms all baselines when $r \leq 100$. However, when $r \leq 300$, DECNws3 outperforms ES and L2O-Swarm and is weaker than DE and CMA-ES. As shown in Table 2, the performance of DECNws3 is worse than DECNws30 and DECNn15. When we use DECNn15 to optimize the complex case, its result is 0.54(0.26), which is better than all baselines.
>
> **Q2:** The protein docking experiments is not very well described. How does this setting turn into continuous optimization and what is the search space?  It would also be helpful to have a consistent set of baselines across all experiments.
>
> **Answer 2:** We have added problem descriptions about protein docking, specifically training and test sets. We parameterize the search space as $\R^{12}$ as in [1]. We also added experimental results for DE, ES, CMA-ES, and Dragonfly. The results are shown below:
> |Methods | 1ATN_7 | 2JEL_1 | 7CEI_1 |
> |- | - | - | - |
> |L2O-Swarm | 2091(25.08) | 2766(24.80) |1690(23.64) |
> |CMA-ES | -6240(100) |-6260(51.8) | -6170(18.4) |
> |ES | -6200(48.1) | -6210(5.05) |-6180(2.47) |
> |DE | -6260(58.1) |-6220(29.2) | -6140(20.5) |
> |Dragonfly | -6160(4.3) |-6120(2.9) | -6103(2.0) |
> |DECNws3 | **-6261(96.71)** |**-6250(84.38)** | **-6193(84.66)** |
>
> [1] Yue Cao and Yang Shen. Bayesian active learning for optimization and uncertainty quantification in protein docking. Journal of chemical theory and computation, 16(8):5334–5347, 2020

---

> ### Comment · Reviewer_g29J · 2022-11-16
> **Comments on Author Response**
>
> I appreciate the authors' thorough response and substantial edits to the draft of the paper. Many of the questions and concerns have been clarified and I am adjusting my score as such. In order to further improve the paper, I think the following would be helpful:
>
> * Some additional proofreading to improve the language (e.g. Section 4.4 should be "Accelerating DECN with GPU")
> * More clarity on how the GPU implementation works and why traditional EAs cannot take advantage from it.
> * A discussion of how DECN contrasts with neuro-evolution method. While it's OK not to include this discussion since they are not directly related, I think it would strengthen the paper to include this.

---

> > ### Author Response · Authors · 2022-11-18
> > **Response to New Comments**
> >
> > Thank you very much for acknowledging our efforts. Your suggestion has improved the quality of our article and given us a deeper understanding of DECN. We also uploaded the modified version.
> >
> > **Q1:** Some additional proofreading to improve the language (e.g. Section 4.4 should be "Accelerating DECN with GPU")
> >
> > **Answer 1:** We will continue to revise the full text.
> >
> > **Q2:** More clarity on how the GPU implementation works and why traditional EAs cannot take advantage from it.
> >
> > **Answer 2:** DECN implements a learnable EA framework based on convolution operations, whose essence is the calculation between matrices. Therefore, we can make use of GPU's parallel processing capability to improve the efficiency of matrix calculation and realize DECN based on PyTorch[1]. There are two reasons why traditional EAs cannot be accelerated using GPU. First of all, there is currently no development framework like PyTorch in EA; Second, the basic crossover, mutation, and selection operations of EAs are challenging to parallelize, which makes it difficult for EAs to benefit from the parallel processing capabilities of GPU. Because EAs are a population-based algorithm, the evolution of multiple populations can have good parallelism. Based on this, researchers can often greatly reduce the actual running time of EA (wall clock), although this often means consuming a lot of computing resources [2, 4].
> >
> > [1] Paszke, Adam, et al. PyTorch: An Imperative Style, High-Performance Deep Learning Library. Advances in Neural Information Processing Systems 32, edited by H. Wallach et al., Curran Associates, Inc., 2019, pp. 8024–35.
> >
> > [2] Salimans, T., Ho, J., Chen, X., Sidor, S., & Sutskever, I. (2017). Evolution strategies as a scalable alternative to reinforcement learning. arXiv preprint arXiv:1703.03864.
> >
> > **Q3:** A discussion of how DECN contrasts with the neuro-evolution method. While it's OK not to include this discussion since they are not directly related. I think it would strengthen the paper to include this.
> >
> > **Answer 3:** We discuss DECN and neuro-evolution methods [3], [4] in our revised version. DECN is an end-to-end optimization framework based on neural networks, while neuro-evolution studies neural network building blocks (such as activation functions), hyperparameters, architectures, and even learning algorithms of neural networks using EAs [3], [4]. In addition, neuro-evolution is largely independent of gradients. DECN is based on a gradient training model and then tested on the target black box problem with no gradient information. In future work, we can use the neuro-evolution method to train deeper DECN for better performance.
> >
> >
> > [3] Stanley, K. O., Clune, J., Lehman, J., & Miikkulainen, R. (2019). Designing neural networks through neuroevolution. Nature Machine Intelligence, 1(1), 24-35.
> >
> > [4] Such, F. P., Madhavan, V., Conti, E., Lehman, J., Stanley, K. O., & Clune, J. (2017). Deep neuroevolution: Genetic algorithms are a competitive alternative for training deep neural networks for reinforcement learning. arXiv preprint arXiv:1712.06567.

---

### Official Review · Reviewer_j4Dd · 2022-10-24

**Confidence:** 4
**Correctness:** 2
**Technical Novelty And Significance:** 3
**Empirical Novelty And Significance:** 3
**Recommendation:** 5

**Clarity, Quality, Novelty And Reproducibility:**

I found part of the writing hard to follow.  For example, the introduction to section 4 was very dense and confusing. As I minor comment it would help to correctly use citations (\citep or \citet depending on the package you are using).  I felt the inspiration from CNNs to be superficial rather than profound.  At least, I needed a lot more convincing that it was profound.  A lot of the statements about the remarkable performance of EAs needed more justification or just needed to be a more balanced assessment of their value.  The method looks very novel and I assume the results can be reproduced.

**Strength And Weaknesses:**

The strength of the approach is that it proposes a novel technique that are shown to have better performance than other well known methods.

For me there are a number of weaknesses.  The approach is motivated by a rather unconvincing (at least to me) analogue between recombination operators and convolutions.  Convolutions in deep neural networks have advantages such as locality and translational invariance that do not seem to carry over to optimisation.  The model of recombination operator is rather strange as it looks like a mixing operator that is likely to result in rapid convergence.  This seems to addressed by a diversity mechanism that is described in a single line.  For me the paper needs to make a more convincing case why the convolution being proposed is in anyway useful.  The resulting model seems to me very complicated and ad hoc with no real motivation.  Maybe I am missing something, but for me I need much more convincing that the proposed method is well motivated.

Of course, the results seem impressive and may well justify the method.  However, the test functions used are a very old set of problems that are often solved efficiently by algorithms that are not generally very competitive.  They are not particularly representative of the type of continuous variable optimisation problem that most people are interested in solving.  To make a convincing empirical case for this approach I would like to see this tested on a modern set of test problems.

**Summary Of The Paper:**

The paper introduces a new evolution based search technique inspired by convolutional networks used for black box optimisation.  The paper introduces a novel mechanism for generating and then selecting individuals in a population based on learnable convolutions.  The paper empirically test the new approach on a number of problems and demonstrates superior performance to some baseline optimisers both in terms of accuracy and speed. They further show that the method benefits from transfer between problems.

**Summary Of The Review:**

The authors have clearly worked hard to develop a novel method to perform black-box optimisation.  However, the field of EAs are full of such novel methods each claiming to be SOTA based on a rather limited empirical test.  For me, I need to see a convincing argument why the approach should work and this seemed missing.  I also need some strong empirical comparison on competitive problems and I felt that despite some effort in this direction it was still not sufficiently convincing.

---

> ### Author Response · Authors · 2022-11-14
> **Response to Reviewer j4Dd (Part 2)**
>
> **Q2:** Convolutions in deep neural networks have advantages such as locality and translational invariance that do not seem to carry over to optimisation.
>
> **Answer 2:** We are dealing with not an image but a population (a collection of solutions). We modify the convolution operation to complete the information interaction between the individuals of the population so as to achieve the purpose of generating a potential solution. We retain the translational invariance property of Convolutions (see Equation 2). However, we did not preserve the locality property of Convolutions. This is because we hope that the information within the population can interact sufficiently to speed up the algorithm's convergence, but the locality property of Convolutions hinders this.
>
> **Q3:** The model of the recombination operator is rather strange as it looks like a mixing operator that is likely to result in rapid convergence. This seems to addressed by a diversity mechanism that is described in a single line.
>
> **Answer 3:** Differential evolution DE [2] can reproduce a unique individual $s^*$ based on $s^*=s_k + \sum_{i=2}^{n-1} F_i(s_i-s_{i+1})$, where $F_i$ is a scaling factor, and $s_k$ is the best solution or is selected from ${s_1, s_2, \cdots, s_n}$.
> After an expression expansion, this process can be summarized by a weighted recombination process with $s_1, s_2, \cdots, s_n$: $s^* = a_1 \times s_1 + a_2 \times s_2 + \cdots +a_n \times s_n = \sum_{i=1}^{n} {a_i \times s_i}$. These operators are manually designed with different parameters ($a_i$).
>
> [2] Swagatam Das and Ponnuthurai Nagaratnam Suganthan. Differential evolution: A survey of the state-of-the-art. IEEE Transactions on Evolutionary Computation, 15(1):4–31, 2010.
>
> **Q4:** Of course, the results seem impressive and may well justify the method. However, the test functions used are a very old set of problems that are often solved efficiently by algorithms that are not generally very competitive. They are not particularly representative of the type of continuous variable optimisation problem that most people are interested in solving. To make a convincing empirical case for this approach I would like to see this tested on a modern set of test problems. I also need some strong empirical comparison on competitive problems and I felt that despite some effort in this direction it was still not sufficiently convincing.
>
> **Answer 4:** In the revised version, we added a real case planar mechanic arm (see Section 4.3). Simple Case (SC): searching for different angles with fixed lengths. Complex Case: searching for different angles and lengths. $gen$ stands for the number of generations for EA baselines. The experimental results are shown as follows:
>
> |Case | $gen$ | $r$ | DE | ES| CMA-ES | L2O-Swarm | DECNws3|
> |- | - | -  | - |- | - | - | -|
> |SC | 10 | 100 | 2.96(1.63) | 11.2(4.70) | 236(46.8) | 40.4(3.89) | **0.42(0.22)**|
> |SC | 10 | 300 | 11.3(14.7) | 45.3(43.3) | 243(125) | 69.5(3.77) | **1.04(1.25)**|
> |SC | 50 | 100 | 1.28(0.60) | 10.7(5.91) | 2.42(0.65) | 40.4(3.89) | **0.42(0.22)**|
> |SC | 50 | 300 | 1.54(0.89) | 42.0(41.0) | 4.06(6.54) | 69.5(3.77) | **1.04(1.25)**|
> |SC | 100 | 100 | 1.20(0.64) | 10.6(5.58) | 1.36(0.35) | 40.4(3.89) | **0.42(0.22)**|
> |SC | 100 | 300 | 1.38(0.71) | 44.9(43.3) | 1.38(0.41) | 69.5(3.77) | **1.04(1.25)**|
> |CC | 100 | 100| 0.81(0.47) | 8.95(6.42) | 0.76(0.20) | 31.9(1.78) | **0.38(0.25)**|
> |CC | 100 | 300 | 6.15(12.2) | 47.8(56.0) | **0.87(0.37)** | 89.1(1.96) | 8.27(21.3) |
>
> In simple cases, DECNws3 outperforms all baselines. Nevertheless, for complex cases, DECNws3 outperforms all baselines when $r \leq 100$. However, when $r \leq 300$, DECNws3 outperforms ES and L2O-Swarm and is weaker than DE and CMA-ES. As shown in Table 2, the performance of DECNws3 is worse than DECNws30 and DECNn15. When we use DECNn15 to optimize the complex case, its result is 0.54(0.26), which is better than all baselines.
>
> **Q5:** I found part of the writing hard to follow. For example, the introduction to section 4 was very dense and confusing.
>
> **Answer 5:** We reorganized the mechanism part of DECN and deleted the redundant part to show the mechanism of DECN more clearly. We urge you to re-evaluate our manuscript and welcome your new comments. Thank you very much!
>
> **Q6:** As I minor comment it would help to correctly use citations (\citep or \citet depending on the package you are using).
>
> **Answer 6:** Fixed.
>
> **Q7:** A lot of the statements about the remarkable performance of EAs needed more justification or just needed to be a more balanced assessment of their value. The method looks very novel, and I assume the results can be reproduced.
>
> **Answer 7:** Thanks very much for your suggestion. We have corrected the statement in the full text.

---

> > ### Comment · Reviewer_j4Dd · 2022-11-15
> > **Quick response**
> >
> > Thank you. I have read your response and will consider it.  I am still a little lost by the motivation of why this approach has a better chance of working than any other.  As you know the literature on optimization is full of "novel approaches", each claiming success.  For me, inventing yet another new method is not science.  The science comes in understanding optimization problems and then coming up with an approach to address some issue (which is what CMA-ES does).  I still don't really get that from your method, but I will reread what you have done.
> >
> > The results you present are more convincing of their utility than the theoretical justification, although I'm not sure what the numbers mean---what is the number in brackets?.  The results are surprisingly different, suggesting that in many cases the baselines just aren't working.  The difficulty I have in interpreting these numbers is to know how much effort has gone into tuning hyper-parameters, etc.  I would also like to see how the results compare with classical methods (Powell's method, or gradient based optimiser if gradients are available)---this would give me some indication of whether the difficulty comes from convergence or the existence of may local optima.  It is useful to know if they were given the same amount of wall time, same number of evaluations, or were all left until they converged. Without this information it is difficult to draw strong conclusions from the table.

---

> > > ### Author Response · Authors · 2022-11-15
> > > **Motivation**
> > >
> > > Thanks for your guide, which gave us a new understanding of the proposed framework. We also uploaded the modified version.
> > >
> > > **Motivation**
> > >
> > > The generalization ability of current evolutionary algorithms (EAs) is poor. Faced with a new black-box optimization task, we need experts to redesign/select the EA's crossover, mutation, and selection operations (including their hyperparameters) to maximize its performance on the target task, resulting in a hand-designed EA with significant application limitation. Most importantly, due to the limitation of expert knowledge, only little target function information is used to assist the design of EA, which makes it challenging to adapt to the target task. How to automatically design optimization strategies according to new tasks is crucial. To the best of our knowledge, there is currently no work to address this issue. We think EA is a generative optimization model that realizes the generation from a random population to an optimal solution by manually designing crossover, mutation, and selection operations. These operations aim to generate potential solutions and retain good ones. The task of automatically designing an optimization strategy is learning how to generate and retain potential solutions automatically. This paper is to show how DECN finish this task.
> > >
> > > By constructing a set of differentiable surrogate functions of the objective black-box function, DECN can allow the designed CRM and SM to learn the strategy of optimizing the objective function. At this point, DECN effectively utilizes the information of the target black-box function to assist the construction of the optimization strategy. The degree of fit of DECN with the target task is much higher than that of the human-designed EA. The following statement may be one-sided: Bayesian optimization also suffers from poor generalization. For example, how to choose/design appropriate acquisition functions for different problems.
> > >
> > > We also found that DECN is a learning-to-optimize framework. Therefore, DECN is compared with the learning-to-optimize framework (L2O-Swarm) for black-box optimization. At the same time, Bayesian optimization is a typical black-box optimization algorithm, so we also compare it with the advanced Dragonfly method.
> > >
> > > **Results**
> > >
> > > The optimized objective function values are shown in the table. The number in brackets is the standard deviation of repeated experiments.
> > > To verify our motivation, DECN is compared with the optimally designed DE, ES, and CMA-ES. The population sizes of DE, ES, CMA-ES, and DECN are 100. DE, ES, and CMA-ES run for 100 generations. For DECNws3, its architecture determines that DECN has only been iterated for three generations. DE, ES, and CMA-ES have 100/3 times as many function evaluations as DECN, which is highly unfair to DECN. Both Dragonfly and L2O-Swarm run to convergence. However, on high-fidelity training datasets and two real-world problems, DECN outperforms these five algorithms. This result validates the motivation of this paper, and there is a great advantage in automatically learning the optimization strategy.
> > >
> > > Especially in Table 1, our scheme outperforms existing methods by a large margin. This is because when training DECN, we use a high-fidelity surrogate function of the target black-box function. The trained DECN contains an optimization strategy that is more tailored to the task. Current DE, ES, CMA-ES, and Dragonfly do not use this information to design their elements. Even if we constantly adjust the hyperparameters of the comparison algorithm, the results are unlikely to be better than DECN. Of course, this is based on the fact that DECN can indeed effectively use this information to learn optimization strategies. This is what DECN is all about. We also test the performance of DECN trained on the low-fidelity surrogate functions, as shown in Table 2. Compared with these five algorithms, deep DECN still has a significant advantage.

---

> > > > ### Comment · Reviewer_j4Dd · 2022-11-21
> > > > **My Current Thinking**
> > > >
> > > > Thank you for your response.  I think your work deserves publishing, but I struggled with the paper in its current form.  As I said,the CNN justification seems confusing, laboured and unconvincing, but the results are impressive.  Building a system that learns the structure of a problem is interesting.  This has always been difficult to do---it can often lead to premature convergence as the problem structure changes as you find better solutions.  I am surprised then you are not comparing to BOA and other algorithms that explicitly try to learn to model the landscape they are optimising.  It is good that your results put your method at a disadvantage because it makes the results a bit more compelling (although I presume each iteration is slower using your method than DE say).  I am still puzzled that the results of your baselines can be two or three orders of magnitudes worst than yours.  This makes me suspicious that you are not running these properly.  It is not my experience that strong baseline algorithms have such a vast discrepancy in performance.  That said, I am not familiar with the problem you are testing on.  I might be prepared to improve my rating slightly.  I think the paper would be much more powerful if it concentrated on the fact that it was learning the problem structure.  If the CNN architecture is necessary then a more compelling explanation of how this is being used in learning the structure of the problem would be helpful (to me it really doesn't look like a CNN and it comes across as poorly motivated).  Finally although there is some evidence of very good performance, I am left with some doubts about the baselines.  If these were addressed the paper would be really strong.

---

> > > > > ### Author Response · Authors · 2022-11-22
> > > > > **Reponse to My Current Thinking (Part 2)**
> > > > >
> > > > > **Q4:** I am surprised then you are not comparing to BOA and other algorithms that explicitly try to learn to model the landscape they are optimising. Finally although there is some evidence of very good performance, I am left with some doubts about the baselines. If these were addressed the paper would be really strong.
> > > > >
> > > > > **Answer 4:** We guess BOA is the Bayesian optimization algorithm. We certainly made a comparison with BOA. This paper compares DECN with Dragonfly, a representative Bayesian optimization algorithm. At the same time, we will also compare with other Bayesian optimization algorithms and surrogate-assisted evolutionary algorithms. Bayesian optimization mainly uses the algorithms in the BoTorch [1] package. The surrogate-assisted evolutionary algorithm mainly adopts the algorithms in the pysamoo package [2]. We will continue to publish the latest experimental results during the open discussion period.
> > > > >
> > > > > In fact, the newly added comparative experiment does not make much sense. DECN guides the design of optimization strategies according to the problem. However, BOA and other schemes for modeling landscapes do not do this. They are just learning the landscape to aid in optimizations, such as creating cheap surrogate functions. However, this does not directly guide the design of optimization strategies, such as selection, crossover, and mutation operations in evolutionary algorithms. DECN achieves this and is more adaptable to the target task.
> > > > >
> > > > > [1] Balandat, M., Karrer, B., Jiang, D., Daulton, S., Letham, B., Wilson, A. G., & Bakshy, E. (2020). BoTorch: a framework for efficient Monte-Carlo Bayesian optimization. Advances in neural information processing systems, 33, 21524-21538.
> > > > > [2] Julian Blank, & Kalyanmoy Deb. (2022). pysamoo: Surrogate-Assisted Multi-Objective Optimization in Python.
> > > > >
> > > > > **Q5:** It is good that your results put your method at a disadvantage because it makes the results a bit more compelling (although I presume each iteration is slower using your method than DE say). I am still puzzled that the results of your baselines can be two or three orders of magnitudes worst than yours. This makes me suspicious that you are not running these properly. It is not my experience that strong baseline algorithms have such a vast discrepancy in performance. That said, I am not familiar with the problem you are testing on.
> > > > >
> > > > > **Answer 5:** We carefully checked our code and found nothing wrong. We were also very surprised by the experimental results presented. For EA baselines, we optimize the combination of the algorithm's crossover, mutation, and selection operations and their corresponding parameters. Note that EA baselines have only been iterated for 100 generations. For the Bayesian optimization baseline (Dragonfly), we also adjusted the module and selected hyperparameters, and the number of function evaluations was consistent with that of DECN. DECN has only been iterated for three generations (3 EMs). EA baselines use 100/3 times as many function evaluations as DECN. Dragonfly and EA baselines do not converge, so their results are relatively poor. The training set of L2O-Swarm is consistent with that of DECN, and the remaining parameters are adjusted to the optimum. However, in Table 1, DECN achieves excellent results with only three iterations. Also, DECN is very fast to optimize the target problem (see Table 5). The experimental results are reasonable in terms of the number of function evaluations.
> > > > >
> > > > > As you state, DECN learns the structure of the problem and inversely guides the design of the optimization strategy. However, BOA and other schemes for modeling landscapes cannot do this. This is the main motivation and advantage of DECN.
> > > > >
> > > > > **You might ask, why not let both Dragonfly and EA baselines run to convergence? Like having them iterate thousands of times.**
> > > > >
> > > > > The higher the number of DECN iterations (the higher the number of EMs), the better the result (see Table 2). **The tradeoff is that the deeper the architecture of DECN, the harder it is to train.** This is a common difficulty faced by many deep architectures. So, it is difficult for DECN to iterate thousands of times, like Bayesian optimization and EAs. This is a shortcoming that we have always wanted to address.

---

> > > > > ### Author Response · Authors · 2022-11-22
> > > > > **Response to My Current Thinking (Part 1)**
> > > > >
> > > > > **Q1:** Thank you for your response. I think your work deserves publishing, but I struggled with the paper in its current form. I might be prepared to improve my rating slightly.
> > > > >
> > > > > **Answer 1: Thank you very much for your support. We always benefit from discussing with you. Your comments have given us a deeper understanding of DECN. Thank you very much for your patient guidance.**
> > > > >
> > > > > **Q2:** As I said,the CNN justification seems confusing, laboured and unconvincing, but the results are impressive. If the CNN architecture is necessary then a more compelling explanation of how this is being used in learning the structure of the problem would be helpful (to me it really doesn't look like a CNN and it comes across as poorly motivated).
> > > > >
> > > > > **Answer 2:** DECN does not use the CNN architecture, and it is difficult to call it a variant of CNN. In the new version, we also do not claim that DECN is an optimization framework based on CNN. It is also difficult for us to imagine how to use CNN to solve optimization problems. DECN is a new optimization framework for dealing with black-box problems. We use the process of an evolutionary algorithm to guide the design of DECN and realize the mapping from a random population to the optimal solution. We need to design a module to ensure the exchange of information between individuals in the population to achieve the function of generating potential solutions. We can achieve this function by modifying the convolution operation accordingly. This is the only similarity with CNN, which all use convolution operations. Meanwhile, to select good individuals to survive, we design the selection module in DECN.
> > > > >
> > > > > **Q3:** Building a system that learns the structure of a problem is interesting. This has always been difficult to do---it can often lead to premature convergence as the problem structure changes as you find better solutions. I think the paper would be much more powerful if it concentrated on the fact that it was learning the problem structure.
> > > > >
> > > > > **Answer 3:** As stated earlier, our motivation is to address the poor generalization ability of current black-box optimization algorithms. Faced with a new black-box optimization task, we need experts to redesign/select their elements, such as EA's crossover, mutation, and selection operations, to maximize their performance on the target task. Since it is difficult for hand-designed operations to utilize the characteristics of the target problem entirely, the adaptability to the target problem is poor. DECN turns optimization strategies into learnable CRMs and SMs. DECN can effectively learn the optimal optimization strategy based on the feedback of the target problem.

---

> > > > > ### Author Response · Authors · 2022-12-11
> > > > > **Did we clarify your comments?**
> > > > >
> > > > > Sorry to bother you. Did our answer clarify your query? We very much welcome your new questions.
> > > > >
> > > > > Best Wishes
> > > > >
> > > > > Authors

---

> ### Author Response · Authors · 2022-11-14
> **Response to Reviewer j4Dd (Part 1)**
>
> Thank you very much for acknowledging our work.
>
> **Q1:** The approach is motivated by a rather unconvincing (at least to me) analogue between recombination operators and convolutions. I felt the inspiration from CNNs to be superficial rather than profound. At least, I needed a lot more convincing that it was profound. The resulting model seems to me very complicated and ad hoc with no real motivation. Maybe I am missing something, but for me I need much more convincing that the proposed method is well motivated.
> For me the paper needs to make a more convincing case why the convolution being proposed is in anyway useful. For me, I need to see a convincing argument why the approach should work and this seemed missing .
>
> **Answer 1:** We design a novel learning-to-optimize framework to handle black-box optimization problems end-to-end. We use the process of an evolutionary algorithm to guide the design of DECN and realize the mapping from a random population to the optimal solution.
>
> First, we need to design a module to ensure the exchange of information between individuals in the population to achieve the function of generating potential solutions (similar to the recombination operators in EA). We can achieve this function by modifying the convolution operation accordingly, which is our motivation for using convolution to design. The designed CRM module achieves this purpose (see Section 3.1).
>
> Second, to survive good individuals for the next layer of DECN, we design the selection module (SM) based on a pairwise comparison between the offspring and input population regarding their fitness (see Section 3.3, Equation 3). We can clearly observe that Equation 3 can indeed keep good individuals.
>
> Third, the untrained DECN does not handle the black-box optimization problem well because it needs information about the target black-box function. In order to better optimize the objective task, we need to design a training set containing objective function information and a practical loss function to guide the parameter training of DECN (see Section 3.5). The characteristics of black-box functions make it difficult for us to obtain their gradient information to assist in the training of DECN. We construct a differentiable surrogate function set of the target black-box function to obtain the information of the target black-box function. The designed loss function is to maximize the difference between the initial population and the output population of DECN to ensure that the initial population is close to the optimal solution.
>
> In Figure 9 of the Appendix, we show the state of the population before and after CRM and SM output, respectively. It is found that DECN is indeed constantly forcing the population to move towards the optimal solution.
>
> There are few learning-to-optimize architectures [1] currently dealing with black-box optimization problems, and their performance is weak. From the experimental results, DECN makes up for the performance disadvantage of the learning-to-optimize architecture in the black-box optimization problem. We also strongly believe that this paper makes an essential contribution to the learning-to-optimize community.
> EA has a unique advantage in dealing with black-box optimization problems. However, its crossover, mutation, and selection modules can only be designed manually based on expert knowledge and cannot effectively interact with the environment (function); that is, they cannot change their elements in real-time to adapt to new problems through the feedback of the objective function. The experimental results show that DECN can indeed solve the problem of EA; at least, it provides a feasible idea to realize the real-time interaction between DECN and the environment and improve performance. The following statement may be partial: We believe that the advantage of deep reinforcement learning over EA lies in its deep architecture and the ability to interact with the environment in real-time to change/optimize its own strategy. DECN is expected to bridge the gap between EA and DRL and take advantage of EA's stable performance advantages to expand to applications suitable for DRL. We consider this paper to be an essential contribution to the evolutionary computation community.
>
> [1] Tianlong Chen, Xiaohan Chen, Wuyang Chen, Howard Heaton, Jialin Liu, Zhangyang Wang, and Wotao Yin. Learning to optimize: A primer and a benchmark. Journal of Machine Learning Research, 23:1–59, 2022.

---

### Official Review · Reviewer_Rucq · 2022-10-25

**Confidence:** 3
**Correctness:** 4
**Technical Novelty And Significance:** 3
**Empirical Novelty And Significance:** 3
**Recommendation:** 6

**Clarity, Quality, Novelty And Reproducibility:**

The quality seems to be very good and the idea seems to be novel.
It doesn't look that the code is available, so the reproducibility is limited, but the general description of experiments is quite detailed.
In general, the clarity is good but some parts could be improved:
- it's not clear how to interpret the results in brackets in tables 1,2,3
- In the case of Dragonfly, there were no results for D=100, it might be good to explain why
- Section 4: it's not sure what is \phi (and the description at the beginning of this section is quite complicated, so maybe could be improved)
- p. 9: Appendix.A.6. -> should be Appendix A.5.

**Strength And Weaknesses:**

Strengths:
- Originality: the idea seems to be quite novel.
- Comprehensive description: many details are provided, and the description of the method and experiments seem to be comprehensive.
- Quality of results: the experiments carried out on unconstrained continuous optimization problems show that DECN surpasses some black-box optimization baselines and obtains good performance when transferred to optimization problems unseen during the training stage. The paper also presents experiments on accelerating DECN using GPU suggesting its good adaptability to GPU acceleration due to the tensor operator.
- The authors are honest about the limitations of DECN.

Weaknesses:
- In some cases, writing could be improved.
- It doesn't look that the code is available, so the reproducibility is limited.

**Summary Of The Paper:**

The paper introduces a deep evolutionary convolution network (DECN) for continuous black-box optimization. DECN is composed of two modules: convolution-based reasoning module (CRM) and selection module (SM). The paper describes both modules and shows how to integrate them. It also contains a description of the process of training including the design of the loss function. The experiments carried out on unconstrained continuous optimization problems show that DECN surpasses some black-box optimization baselines and obtains good performance when transferred to optimization problems unseen during the training stage. The paper also presents experiments on accelerating DECN using GPU suggesting its good adaptability to GPU acceleration due to the tensor operator. The main article is also followed by the Appendix providing more details, including limitations of the model.

**Summary Of The Review:**

In general, I like the paper and the idea - it seems to be quite original and the quality of the results seems to be very good. The weaknesses of the article that I see are related mostly to clarity and lack of reproducibility. The weaknesses of the method are clearly explained in the appendix. The quality of the results seems to be very good.

---

> ### Author Response · Authors · 2022-11-14
> **Response to Reviewer Rucq**
>
> Thank you very much for acknowledging our work. We finally find our soulmate, and you understand what we are doing. Thank you again.
>
> **Q1:** It doesn't look that the code is available, so the reproducibility is limited.
>
> **Answer 1:** We also provide implementation code for Pytorch version of DECN (see Supplementary Materials).
>
> **Q2:** In general, the clarity is good but some parts could be improved.
>
> **Answer 2:** Thank you very much for your valuable comment. We have rewritten the mechanistic and experimental sections of the article to address your doubts. We may need to trouble you to re-review our revised version. We are really sorry for causing you trouble.
>
> **Q3:** it's not clear how to interpret the results in brackets in tables 1,2,3.
>
> **Answer 3:** \*(\*) represents the mean and standard deviation of repeated experiments.
>
> **Q4:** In the case of Dragonfly, there were no results for D=100, it might be good to explain why.
>
> **Answer 4:** When D=100, our scheme can be completed in 1s, but Dragonfly takes dozens of hours, and the quality of the solution is not high, so we do not show its results.
>
> **Q5:** Section 4: it's not sure what is \phi (and the description at the beginning of this section is quite complicated, so maybe could be improved)
>
> **Answer 5:** We removed \phi because not only did it have no effect, but it added complexity to the description. Specifically, “the procedure of DECN can be formulated as $S_t=G_{\theta} (S_0, f(s|\xi))$, where $s$ is the target to be optimized, and $\xi$ is the known parameters of $f$.} Based on $\theta$, DECN optimizes $f(s|\xi)$ by $S_t=G_{\theta} (S_0, f(s|\xi))$. To be noted, $\theta$ is the parameters (strategies) of $G$, while $G$ is an abstract function remarking the optimization process, where $S_0$ is the initial population and $S_t$ is the output population.”
>
> **Q6:** p. 9: Appendix.A.6. -> should be Appendix A.5.
>
> **Answer 6:** Fixed.

---

> ### Author Response · Authors · 2022-12-06
> **Did we clarity your comments? (2)**
>
> Sorry for bothering you.
>
> If possible, we hope that you will take the time out of your busy schedule to re-evaluate the revised paper. Thank you very much.
>
> Best Wishes
>
> Authors

---

> > ### Author Response · Authors · 2022-12-11
> > **Sorry to bother you**
> >
> > Sorry to bother you. Did our answer clarify your query? We very much welcome your new questions.
> >
> > Best Wishes
> >
> > Authors

---

> > > ### Comment · Reviewer_Rucq · 2022-12-11
> > > **The updated version**
> > >
> > > I've read the updated version of the paper. The changes are significant. I appreciate addressing the issues I raised, especially adding the statement regarding the reproducibility of the results. I've reviewed the issues raised by other reviewers and the authors' responses. I am still leaning toward accepting the paper. However, I also have suggestions for further improvements:
> > > - I have some doubts regarding the caption under Figure 3: f(s = {x1, x2}, xi = {0, 0}) =(x1 − 0)^2(x2 − 0)^2, xi ∈ [−1, 1]. Shouldn't it be (x1 − 0)^2 + (x2 − 0)^2 ? Also, I am not sure how to interpret xi = {0, 0}, i.e. what is xi in this context?
> > > - In the description of Algorithm 1 we have "Update θ by minimizing −1/m SUM_j L_j" - it seems that L_j is negative and we would rather like to minimize the SUM of L_j, so shouldn't be it 1/m SUM_j L_j instead of -1/m SUM_j L_j ?
> > > - Fig. 4: best solutions as well as S1', S2', S14' are not visible, so probably it would be good to improve the figures. Also, for clarity, it might be good to improve the caption "Visualization of the optimization process in consecutive generations."

---

### Official Review · Reviewer_9TVi · 2022-10-25

**Confidence:** 3
**Correctness:** 2
**Technical Novelty And Significance:** 2
**Empirical Novelty And Significance:** 2
**Recommendation:** 5

**Clarity, Quality, Novelty And Reproducibility:**

- Parts of the paper are not clear, especially the notation in Section 4. The notation in other sections are also not very clear.
- The quality of the paper could be significantly improved. First, the motivation for modeling the operations through convolutional networks is not clear. Parts of the exposition are not clear, both in terms of notation and grammar.
- I did not check, but code was provided for reproducing the proposed method.

**Strength And Weaknesses:**

Strengths:
1. Adopting neural networks within evolutionary algorithms for black-box optimization is interesting.
2. The method seems to be effective, especially for the protein docking problem, task-specific optimization methods have not been compared against.

Weaknesses:
1. While the idea of modeling the recombination and selection operators through neural networks .
2. The approach may suffer from the common problems with current learning based methods, e.g., overfitting, lack of domain generalization etc. If the optimization landscapes used for training and deployment are not similar, the approach may lead to sub-optimal solutions. Furthermore, there is no mechanism to identify such scenarios. The paper acknowledge this limitation.

Other Comments:
1. The topic of the paper, evolutionary black-box optimization, may be more suitable and better appreciated by evolutionary optimization community.

**Summary Of The Paper:**

This paper proposes a black-box optimization algorithm by incorporating convolutional neural networks into an evolutionary algorithm. Convolutional layers are used for both generating the offspring as well as selecting the offsprings that survive. Experiments are performed on standard black-box functions and protein docking.

**Summary Of The Review:**

The paper proposes to incorporate convolutional networks into evolutionary algorithms for solving black-box optimization problems. Both recombination and selection operations are modeled through learned convolutional networks. The motivation for modeling the operators through convolutional layers is not well justified, given that the population members do not necessarily have spatial relations. Experimental evaluation is limited in some respects.

Post Rebuttal Update:

I read the response of the authors, the discussion between reviewers and authors, and the updated paper.

The original submission certainly had some writing and presentation issues. The author's rebuttal pointed to at least two reviewers who misunderstood the paper. The paper has since been revised significantly. I think the revised form is certainly better, but there are still some claims that are too strong, and I do not agree with them. For example, the claim that evolutionary algorithms have poor generalization.

I have slightly increased my rating, but I still lean toward rejection. I feel the paper has potential but needs further revision.

---

> ### Author Response · Authors · 2022-11-14
> **Response to Reviewer 9TVi (Part 2)**
>
> **Q2:** Experimental evaluation is limited in some respects.
>
> **Answer 2:** In the revised version, we also added a real case planar mechanic arm (see Section 4.3). Simple Case (SC): searching for different angles with fixed lengths. Complex Case: searching for different angles and lengths. $gen$ stands for the number of generations for EA baselines. The experimental results are shown as follows:
>
> |Case | $gen$ | $r$ | DE | ES| CMA-ES | L2O-Swarm | DECNws3|
> |- | - | -  | - |- | - | - | -|
> |SC | 10 | 100 | 2.96(1.63) | 11.2(4.70) | 236(46.8) | 40.4(3.89) | **0.42(0.22)**|
> |SC | 10 | 300 | 11.3(14.7) | 45.3(43.3) | 243(125) | 69.5(3.77) | **1.04(1.25)**|
> |SC | 50 | 100 | 1.28(0.60) | 10.7(5.91) | 2.42(0.65) | 40.4(3.89) | **0.42(0.22)**|
> |SC | 50 | 300 | 1.54(0.89) | 42.0(41.0) | 4.06(6.54) | 69.5(3.77) | **1.04(1.25)**|
> |SC | 100 | 100 | 1.20(0.64) | 10.6(5.58) | 1.36(0.35) | 40.4(3.89) | **0.42(0.22)**|
> |SC | 100 | 300 | 1.38(0.71) | 44.9(43.3) | 1.38(0.41) | 69.5(3.77) | **1.04(1.25)**|
> |CC | 100 | 100| 0.81(0.47) | 8.95(6.42) | 0.76(0.20) | 31.9(1.78) | **0.38(0.25)**|
> |CC | 100 | 300 | 6.15(12.2) | 47.8(56.0) | **0.87(0.37)** | 89.1(1.96) | 8.27(21.3) |
>
> In simple cases, DECNws3 outperforms all baselines. Nevertheless, for complex cases, DECNws3 outperforms all baselines when $r \leq 100$. However, when $r \leq 300$, DECNws3 outperforms ES and L2O-Swarm and is weaker than DE and CMA-ES. As shown in Table 2, the performance of DECNws3 is worse than DECNws30 and DECNn15. When we use DECNn15 to optimize the complex case, its result is 0.54(0.26), which is better than all baselines.
>
> **Q3:** The approach may suffer from the common problems with current learning based methods, e.g., overfitting, lack of domain generalization etc. If the optimization landscapes used for training and deployment are not similar, the approach may lead to sub-optimal solutions. Furthermore, there is no mechanism to identify such scenarios. The paper acknowledge this limitation.
>
> **Answer 3:** In Section 4.1, the experiment of Results on Low-fidelity Training Dataset is to verify the domain generalization of DECN. The F1-F3 used in the training set is very different from the objective function and is a low-fidelity training set. However, the experimental results show that DECN still has a certain domain generalization ability. At the same time, when the number of DECN layers is deep and there is no weight sharing, the domain generalization ability is strong. Our statement below may be one-sided: our scheme enables the design of pre-trained models for black-box optimization. We can use a large number of black-box optimization problems as a training set and use the meta-learning strategy to train DECN to obtain a pre-trained model with high domain generalization ability. However, the focus on certain types of problems is poor. We can refine the pre-trained model for a specific black-box optimization problem for better performance.
>
> **Q4:** Parts of the paper are not clear, especially the notation in Section 4. The notation in other sections are also not very clear.
>
> **Answer 4:** We have carefully checked the full text to avoid such errors. For details, see the blue section of the revised paper. We need to trouble you to re-review our revised version. We are really sorry for causing you trouble.

---

> ### Author Response · Authors · 2022-11-14
> **Response to Reviewer 9TVi (Part 1)**
>
> **Q1:** Summary Of The Review: The paper proposes to incorporate convolutional networks into evolutionary algorithms for solving black-box optimization problems. Both recombination and selection operations are modeled through learned convolutional networks. The motivation for modeling the operators through convolutional layers is not well justified, given the at the population members do not necessarily have spatial relations.
> Summary Of The Paper: This paper proposes a black-box optimization algorithm by incorporating convolutional neural networks into an evolutionary algorithm. Convolutional layers are used for both generating the offspring as well as selecting the offsprings that survive.
>
> **Answer 1:** Thanks a lot for your comment. Your comments have made a qualitative change in the clarity of the paper.
>
> Based on your summary, we believe you have misunderstood this article.
>
> Our understanding of yours is summarized as follows: This paper uses convolutional neural networks to act as a recombination operator in an evolutionary algorithm to improve the performance of evolutionary algorithms. Maybe you understand the algorithm flow is: input the initial population into the EA modified with DECN and output the final solution.
>
> Our practical contribution is: that black-box optimization is a fundamental problem. We design a novel learning-to-optimize framework to handle black-box optimization problems end-to-end. The initial population is input into DECN, and DECN outputs the final solution. Our framework is not embedded in evolutionary algorithms. We also did not replace the evolutionary algorithm's recombination operators with the designed modules. We use the process of an evolutionary algorithm to guide the design of DECN and realize the mapping from a random population to the optimal solution. We need to design a module to ensure the exchange of information between individuals in the population to achieve the function of generating potential solutions (similar to the recombination operators in EA). We can achieve this function by modifying the convolution operation accordingly, which is our motivation for using convolution to design. Meanwhile, to select good individuals, we design the selection module in DECN.
>
> **Motivation 1:**
>
> The adaptability of current evolutionary algorithms (EAs) is poor. Faced with a new black-box optimization task, we need experts to redesign/select the EA's crossover, mutation, and selection operations (including their hyperparameters) to maximize its performance on the target task, resulting in a hand-designed EA with significant application limitation. Most importantly, due to the limitation of expert knowledge, only little target function information is used to assist the design of EA, which makes it challenging to adapt to the target task. How to automatically design optimization strategies according to new tasks is crucial. To the best of our knowledge, there is currently no work to address this issue. We think EA is a generative optimization model that realizes the generation from a random population to an optimal solution by manually designing crossover, mutation, and selection operations. These operations aim to generate potential solutions and retain good ones. The task of automatically designing an optimization strategy is learning how to generate and retain potential solutions automatically. This paper is to show how DECN finish this task.
>
> By constructing a set of differentiable surrogate functions of the objective black-box function, DECN can allow the designed CRM and SM to learn the strategy of optimizing the objective function. At this point, DECN effectively utilizes the information of the target black-box function to assist the construction of the optimization strategy. The degree of fit of DECN with the target task is much higher than that of the human-designed EA.
>
> **Motivation 2:**
>
> There are few learning-to-optimize architectures [1] currently dealing with black-box optimization problems, and their performance is weak. From the experimental results, DECN makes up for the performance disadvantage of the learning-to-optimize architecture in the black-box optimization problem. We also strongly believe that this paper makes an essential contribution to the learning-to-optimize community.
>
> [1] Tianlong Chen, et al. Learning to optimize: A primer and a benchmark. JMLR, 23:1–59, 2022.

---

> ### Author Response · Authors · 2022-12-06
> **Did we clarity your comments? (2)**
>
> Sorry for bothering you.
>
> If possible, we hope that you will take the time out of your busy schedule to re-evaluate the revised paper. Thank you very much.
>
> Best Wishes
>
> Authors

---

> ### Author Response · Authors · 2022-12-11
> **EAs have poor adaptability.**
>
> Thank you very much.
>
> We are very sorry that our clerical error has caused you trouble. "generalization" should be "adaptability".
>
> This part is modified as follows: Although the generalization ability of EAs is good, the no-free lunch theorem [1] tells us that we need to specialize an algorithm to a specific problem. Faced with a new black-box task, we need experts to redesign the EA’s operations to maximize its performance on the target task, resulting in poor adaptability.
>
> Best Wishes
>
> Authors

---

### Official Review · Reviewer_Tk68 · 2022-10-26

**Confidence:** 4
**Correctness:** 2
**Technical Novelty And Significance:** 2
**Empirical Novelty And Significance:** 2
**Recommendation:** 3

**Clarity, Quality, Novelty And Reproducibility:**

The overall paper is not always clear, the writing is lacking at times and is not very precise. For example, the DECN model presentation at the beginning of sec. 4 is not very helpful nor clear, and I am not sure what we want to express with figure 1. Likewise for the explanations in sec. 4.1 on the Convolution Reasoning Module, the explanations are long and not always necessary to the point, informative/pedagogical or even relevant. I found it difficult to extract the useful information from the method over the sections. Another example of clarity lacking is figure 7, which is hard to read and not very helpful regarding the proposal.

Other examples of unclear statements are with the beginning of sec. 5.1, which states an important methodological element as “We test DECN on six standard black-box functions (Appendix Table 6). We train DECNs on the original functions, and then we train six DECNs in this part.” I don’t get it, are the six standard black-box functions directly related to the six DECNs trained in that part? And is the training made on the original functions (which concept is not properly defined) done over all these functions simultaneously, or one at the time? Said otherwise, do we have a DECN trained on all original functions, or six DECNs because we have six original functions (I know from the appendix this is not the case since there are only three original functions, F1, F2 and F3).

The overall quality of the experiments remains quite below the usual standards. It is written « The parameters of DE, ES, CMA-ES, and Dragonfly are adjusted to be optimal. », which is not specific. Are these parameters adjusted once, or for each problem? I don’t get the transferability experiments (sec. 5.3), I am not even sure what we mean here by transferability. The lack of details over the experiments makes reproducibility difficult to achieve in my opinion.

I am really unsure about the speedup presented by using GPUs over other approaches. First, the execution time with the EA is really high, higher than what I see and know with these optimization approaches. There might be implementation or configuration issues – which library has been used to run them. Also, in real-life, more of the computation is usually passed on fitness evaluation. where GPUs are often not helpful.

As for the originality, this is not the first work on learning-to-optimize and the overall motivations are not well developed, not in a convincing way. I am far from being convinced by the approach, it doesn’t sound like the way to go for learning-to-optimize, as the operations are based on fitness values and fitness ranking, that signal appears rather weak for guiding the recombination operator.


**Strength And Weaknesses:**

Strength
- The proposal is relatively straightforward and carries interesting ideas for the  learning-to-optimize context.
- The results show the approach is able to outperform some black-box optimization baselines for the problems tested.
- The approach appears to be GPU-friendly.

Weaknesses
- The approach proposed relies on some intuitive idea that convolutions in neural networks make some computations that can be useful to replace combination operators in an evolutionary algorithm. This intuition appears really rough to me, and the overall conceptual proposal is not well principled.
- The writing clarity of the paper is low, there is a lack of precision and details, several elements remain unclear and reproducibility appears not obvious to me.
- Problems tackled are a subset of what can be tested for real-valued black-box optimization (although they are the common baselines).


**Summary Of The Paper:**

The paper proposes the use of convolutional neural networks to act as a recombination operator in a population-based black-box optimization algorithm (aka evolutionary algorithm). That operator is combining the solutions making the population, ranked according to their fitness, with weights provided by the learned convolutions making the neural network. The recombination is made both between individuals, through some kind of mix-up between all individuals of the population through a depthwise separable convolution. The weights are modulated according to the fitness of the individuals used, and multiple convolutions are averaged to produce the output individuals. The neural network model is trained on some problems to generate a combination operator to use for similar problems. Results are provided by comparing the proposed approach with some evolutionary algorithms, with competitive performances.

**Summary Of The Review:**

The overall proposal is not well motivated, the explanations are not sufficiently precise and clear, the experiments reported appear good, but still do not bring high confidence in them given the lack of clarity and precision of the experimental configuration.

---

> ### Author Response · Authors · 2022-11-14
> **Response to Reviewer Tk68 (Part 4)**
>
> **Q7:** I don’t get the transferability experiments (sec. 5.3), I am not even sure what we mean here by transferability.
>
> **Answer 7:** We modify this part of the experiment as follows:
>
> **Results on Low-fidelity Training Dataset**.
> The training of DECN requires a differentiable surrogate function for the black-box optimization problem. However, accurate high-fidelity surrogate functions are difficult to obtain. Therefore, this section tests the performance of DECNs trained on low-fidelity surrogate functions. Three functions in Table 6 are employed as the low-fidelity surrogate functions for each function in Table 7. Here, the whole functions in Table 6 are employed as $F^{train}$ in order to train one DECN, and then the results on each function of Table 7 are shown in Table 2. For example, we show the designed training and testing datasets for the F4 function as follows: $F^{train} = \{F1(s|\xi_{1,i}), F2(s|\xi_{2,i}), F3(s|\xi_{3,i})\}, \  \  F^{test} = \{F4(s|\xi^{test})\} $
> Meanwhile, we also test the impact of different architectures on DECN, including the different number of layers and whether weights are shared between layers. We design three models, including DECNws3, DECNn15, and DECNws30. DECNn15 does not share parameters across different EMs. DECNn15 does not share parameters across 15 EMs.
>
> **Q8:** I am really unsure about the speedup presented by using GPUs over other approaches. First, the execution time with the EA is really high, higher than what I see and know with these optimization approaches. There might be implementation or configuration issues – which library has been used to run them. Also, in real-life, more of the computation is usually passed on fitness evaluation. where GPUs are often not helpful.
>
> **Answer 8:** All methods are tested in a unified environment, and EA uses the Geatpy package. All experimental studies are performed on a Linux PC with Intel Core i7-10700K CPU at 3.80GHz and 32GB RAM.
>
> The running time of EA is higher than expected because the population size is relatively large, and the population size is K\*L\*L, where K=32, L=10 or 80. We simulate all cases in a unified environment. Thus, the time of function evaluation consumed by DECN and EA is basically the same. The GPU can help speed up the CRM and SM modules. However, GPU cannot accelerate EA's crossover, mutation, and selection modules. In the case of a large population of individuals, these operations have a high time consumption. Therefore, our results are reasonable. We understand your concerns, although GPUs have no boost to function estimates. However, you cannot deny that DECN accelerates the process of generating and selecting solutions, which is also our main point. Furthermore, we do not claim that the GPU has an augmentation effect on the function estimation of DECN.
>
> **Q9:** I am far from being convinced by the approach, it doesn’t sound like the way to go for learning-to-optimize, as the operations are based on fitness values and fitness ranking, that signal appears rather weak for guiding the recombination operator.
>
> **Answer 9:** Reference [1] defines L2O as follows: Learning to optimize (L2O) is an emerging approach that leverages machine learning to develop optimization methods, aiming at reducing the laborious iterations of hand engineering. Our scheme DECN revised convolution operations to design a new black-box optimization scheme. DECN to realize the move from hand-designed searching strategies to learned searching strategies in population-based optimization. Our scheme is a model-free learning-to-optimize scheme.
> You may be familiar with learning-to-optimize frameworks that deal with differentiable objectives, where gradient information for the optimization objective is available. However, the object we deal with is a black-box optimization problem whose gradient information cannot be obtained, and the optimization can only be guided by fitness values.

---

> ### Author Response · Authors · 2022-11-14
> **Response to Reviewer Tk68 (Part 3)**
>
> **Q6:** Problems tackled are a subset of what can be tested for real-valued black-box optimization (although they are the common baselines).
>
> **Answer 6:** In the revised version, we added a real case planar mechanic arm (see Section 4.3). Simple Case (SC): searching for different angles with fixed lengths. Complex Case: searching for different angles and lengths. $gen$ stands for the number of generations for EA baselines. The experimental results are shown as follows:
>
> |Case | $gen$ | $r$ | DE | ES| CMA-ES | L2O-Swarm | DECNws3|
> |- | - | -  | - |- | - | - | -|
> |SC | 10 | 100 | 2.96(1.63) | 11.2(4.70) | 236(46.8) | 40.4(3.89) | **0.42(0.22)**|
> |SC | 10 | 300 | 11.3(14.7) | 45.3(43.3) | 243(125) | 69.5(3.77) | **1.04(1.25)**|
> |SC | 50 | 100 | 1.28(0.60) | 10.7(5.91) | 2.42(0.65) | 40.4(3.89) | **0.42(0.22)**|
> |SC | 50 | 300 | 1.54(0.89) | 42.0(41.0) | 4.06(6.54) | 69.5(3.77) | **1.04(1.25)**|
> |SC | 100 | 100 | 1.20(0.64) | 10.6(5.58) | 1.36(0.35) | 40.4(3.89) | **0.42(0.22)**|
> |SC | 100 | 300 | 1.38(0.71) | 44.9(43.3) | 1.38(0.41) | 69.5(3.77) | **1.04(1.25)**|
> |CC | 100 | 100| 0.81(0.47) | 8.95(6.42) | 0.76(0.20) | 31.9(1.78) | **0.38(0.25)**|
> |CC | 100 | 300 | 6.15(12.2) | 47.8(56.0) | **0.87(0.37)** | 89.1(1.96) | 8.27(21.3) |
>
> In simple cases, DECNws3 outperforms all baselines. Nevertheless, for complex cases, DECNws3 outperforms all baselines when $r \leq 100$. However, when $r \leq 300$, DECNws3 outperforms ES and L2O-Swarm and is weaker than DE and CMA-ES. As shown in Table 2, the performance of DECNws3 is worse than DECNws30 and DECNn15. When we use DECNn15 to optimize the complex case, its result is 0.54(0.26), which is better than all baselines.

---

> ### Author Response · Authors · 2022-11-14
> **Response to Reviewer Tk68 (Part 2)**
>
> **Q3:** The overall paper is not always clear, the writing is lacking at times and is not very precise. For example, the DECN model presentation at the beginning of sec. 4 is not very helpful nor clear, and I am not sure what we want to express with figure 1. Likewise for the explanations in sec. 4.1 on the Convolution Reasoning Module, the explanations are long and not always necessary to the point, informative/pedagogical or even relevant. I found it difficult to extract the useful information from the method over the sections. Another example of clarity lacking is figure 7, which is hard to read and not very helpful regarding the proposal.
>
> **Answer 3:** Your comments are excellent, and you are also welcome to check our revised version. We reorganized the mechanism part of DECN, removed redundant parts, and only showed the core steps of DECN (see Section 3) so that readers can follow the process of DECN clearly. At the same time, we put the background part in Appendix. Figure 1 expresses the optimization goal of DECN: gradually force the initialization population to move near the optimal solution. We have deleted Figure 1. In Figure 7, we give an example to show how DECN works. We have carefully considered your comments, and we still think that Figure 7 is helpful for readers to understand DECN, so we keep it in the main text.
>
> **Q4:** Other examples of unclear statements are with the beginning of sec. 5.1, which states an important methodological element as “We test DECN on six standard black-box functions (Appendix Table 6). We train DECNs on the original functions, and then we train six DECNs in this part.” I don’t get it, are the six standard black-box functions directly related to the six DECNs trained in that part? And is the training made on the original functions (which concept is not properly defined) done over all these functions simultaneously, or one at the time? Said otherwise, do we have a DECN trained on all original functions, or six DECNs because we have six original functions (I know from the Appendix this is not the case since there are only three original functions, F1, F2 and F3).
>
> **Answer 4:** We are very sorry for the inconvenience caused to you. We reorganized this section and gave specific definitions of training and test sets (see Section 4.1). The details are shown as follows:
> \paragraph{Results on High-fidelity Training Dataset}
> For each function in Appendix Table 7, we produce the training dataset as follows: 1) Randomly initialize the input population $S_0$; 2) Randomly produce a shifted objective function $f_i(s|\xi)$ by adjusting the corresponding location of optima-namely, adjusting the parameter $\xi$; 3) Evaluate $S_0$ by $f_i(s|\xi)$; 4) Repeat Steps 1)-3) to generate the corresponding dataset. We show the designed training and testing datasets as follows:
> $$
> F^{train} = \{F4(s|\xi_{1}^{train}),\cdots, F4(s|\xi_{m}^{train})\},
> F^{test} = \{F4(s|\xi^{test})\}
> $$
> $F^{train}$ and $F^{test}$ are comprised of the same essential function but vary in the location of optima obtained by setting different combinations of $\xi$ (called $b_i$ in Table 7). $F^{train}$ can be considered as the high-fidelity surrogate functions of $F^{test}$. We train DECN on $F^{train}$, and then we test the performance of DECN upon $F^{test}$, where the values of $\xi^{test}$ not appearing in the training process.
>
> At the same time, we rewrote the experimental section. The main revisions are
> 1.	clearly give the purpose of each experiment;
> 2.	clearly define the training set and test set for each question;
> 3.	set up a complete experiment.
> Please move to Section 4 of the modified version to review whether our modification is appropriate.
>
> **Q5:** The overall quality of the experiments remains quite below the usual standards. It is written « The parameters of DE, ES, CMA-ES, and Dragonfly are adjusted to be optimal. », which is not specific. Are these parameters adjusted once, or for each problem? The lack of details over the experiments makes reproducibility difficult to achieve in my opinion.
>
> **Answer 5:** DECN is compared with standard EA baselines (DE(DE/rand/1/bin), ES(($\mu$,$\lambda$)-ES), and CMA-ES), L2O-swarm (a representative L2O method for black-box optimization), and Dragonfly (the state-of-the-art Bayesian optimization). DE and ES are implemented based on geatpy [2], and CMA-ES is implemented by pymoo [3]. These parameters are adjusted for each problem. We have added parameters for the compared algorithms and descriptions of different experimental setups to make them reproducible. We also provide implementation code for Pytorch version of DECN (see Supplementary Materials).
>
> [2] Jazzbin et.al. geatpy: The genetic and evolutionary algorithm toolbox with high performance in python, 2020. http://www.geatpy.com/
>
> [3] J. Blank and K. Deb. pymoo: Multi-objective optimization in python. IEEE Access, 8:89497–89509, 2020

---

> ### Author Response · Authors · 2022-11-14
> **Response to Reviewer Tk68 (Part 1)**
>
> **Q1:** The overall proposal is not well motivated, the explanations are not sufficiently precise and clear, the experiments reported appear good, but still do not bring high confidence in them given the lack of clarity and precision of the experimental configuration.
>
> **Answer 1:** 1) We have reorganized the motivation of this article (see answer 2).
> 2) We reorganized the mechanism part of DECN and deleted the redundant part to show the mechanism of DECN more clearly.
> 3) We rewrote the experimental section to clearly state the experimental setup to ensure the reproducibility of the experiment. At the same time, we also submitted the implementation code of Pytorch version of DECN (see Supplementary Materials).
> We urge you to re-evaluate our manuscript and welcome your new comments. Thanks!
>
> **Q2:** The approach proposed relies on some intuitive idea that convolutions in neural networks make some computations that can be useful to replace combination operators in an evolutionary algorithm. This intuition appears really rough to me, and the overall conceptual proposal is not well principled. As for the originality, this is not the first work on learning-to-optimize and the overall motivations are not well developed, not in a convincing way.
>
> **Answer 2:** Thanks a lot for your comment. Your comments have made a qualitative change in the clarity of the paper.
>
> Based on your summary, we believe you have misunderstood this article.
>
> Our understanding of yours is summarized as follows: This paper uses convolutional neural networks to act as a recombination operator in an evolutionary algorithm to improve the performance of evolutionary algorithms. Maybe you understand the algorithm flow is: input the initial population into the EA modified with DECN and output the final solution.
>
> Our practical contribution is: the black-box optimization is a fundamental problem. We design a novel learning-to-optimize framework to handle black-box optimization problems end-to-end. The initial population is input into DECN, and DECN outputs the final solution. Our framework is not embedded in evolutionary algorithms. We also did not replace the evolutionary algorithm's recombination operators with the designed modules. We use the process of an evolutionary algorithm to guide the design of DECN and realize the mapping from a random population to the optimal solution. We need to design a module to ensure the exchange of information between individuals in the population to achieve the function of generating potential solutions (similar to the recombination operators in EA). We can achieve this function by modifying the convolution operation accordingly, which is our motivation for using convolution to design. Meanwhile, to select good individuals to survive, we design the selection module in DECN.
>
> **Motivation 1:**
>
> The generalization ability of current evolutionary algorithms (EAs) is poor. Faced with a new black-box optimization task, we need experts to redesign/select the EA's crossover, mutation, and selection operations (including their hyperparameters) to maximize its performance on the target task, resulting in a hand-designed EA with significant application limitation. Most importantly, due to the limitation of expert knowledge, only little target function information is used to assist the design of EA, which makes it challenging to adapt to the target task. How to automatically design optimization strategies according to new tasks is crucial. To the best of our knowledge, there is currently no work to address this issue. We think EA is a generative optimization model that realizes the generation from a random population to an optimal solution by manually designing crossover, mutation, and selection operations. These operations aim to generate potential solutions and retain good ones. The task of automatically designing an optimization strategy is learning how to generate and retain potential solutions automatically. This paper is to show how DECN finish this task.
>
> By constructing a set of differentiable surrogate functions of the objective black-box function, DECN can allow the designed CRM and SM to learn the strategy of optimizing the objective function. At this point, DECN effectively utilizes the information of the target black-box function to assist the construction of the optimization strategy. The degree of fit of DECN with the target task is much higher than that of the human-designed EA.
>
> **Motivation 2:**
>
> There are few learning-to-optimize architectures [1] currently dealing with black-box optimization problems, and their performance is weak. From the experimental results, DECN makes up for the performance disadvantage of the learning-to-optimize architecture in the black-box optimization problem. We also strongly believe that this paper makes an essential contribution to the learning-to-optimize community.
>
> [1] Tianlong Chen, et al. Learning to optimize: A primer and a benchmark. JMLR, 23:1–59, 2022.

---

> ### Comment · Reviewer_Tk68 · 2022-12-02
> **Update of my evaluation following responses and paper update**
>
> I read all replies from the authors to the reviews and the exchanges that followed with the reviewer. I also had a reading of the updated version of the document. I recognize the efforts the authors put into answering the reviewers and making a major update to their paper. However, following this, I am not changing my evaluation, some points are even strenghtening my confidence in it. I don't think the paper is satisfactory state in its current state, I think that there are some fundamental issues in it that make it not suitable for publication.
>
> - There is an argument that "The generalization ability of EAs is poor. Faced with a new black-box optimization task, we need experts to redesign the EA’s crossover, mutation, and selection operations to maximize its performance on the target task, resulting in a hand-designed EA with big application limitation." (second paragraph of the updated paper). As an experimented researcher in EA, I strongly disagree with this. It is not true that with EA, experts are redesigning their crossover, mutation and selection operators for the application at hand, quite the opposite. EAs are meta-heuristics known to provide "good enough" to a given problem, with no customization required. See for example the algorithm CMA-ES, which is known to be among the best algorithm for black-box numerical optimization, it is quite unusual to customize this algorithm for any specific use, it is working out-of-the-box in most cases. Of course, some hyper-parameter adjustments may be helpful, like it is with most ML algorithms. Of course, specializing an algorithm to a specific problem may provides some gains at the expense of wide applicability (remember the no free lunch theorem), but that's not something that commonly done when using EAs.
> - In fact, I think that the point the authors want to bring is the capacity to develop an algorithm able to learn automatically the right optimization scheme for a given family of problem. That's the definition of learning to optimize (L2O) and it is a fine goal by itself, but I think it is deceiving to claim that generic black-box algorithms are not able to learn or are not generalizing. Several of them (e.g. genetic algorithm) are not designed to learn but are still meta-heuristics, that can be applied to a wide variety of problems -- they are generalizing, although they are not able to specialize that much to a given context. Others, like CMA-ES with its adaptation of a covariance matrix, is able to learn when executed on a given optimization problem, although it is not learning across several runs. Let's compare apples with apples and do not claim that black-box optimization algorithms are not able to do L2O when they are not designed for that.
> - In my review, I stated that the convolutions are used in replacement of the variation operators (crossover and mutation), which appears to raise a reaction from the authors. I still maintain this is the case, the proposed approach of modeled against the generic EA iterative optimization model (let's call it population-based optimization if the evolutionary term appears shocking or inexact), where a population of candidate solutions is modified (variation operator) and resampled according to their performance (selection) over many iterations (generations) until convergence. The learned convolutions proposed in the paper is simply a way to learn the variation operator.
> - The justification of using convolution and using an L x L structure is weak, the underlying principles relying on convolution are not clear to me. Proposing a yet new optimization approach is not very interesting as far as I am concerned, but proposing new, sound principles for black-box optimization would be. I would have liked to get better articulated explanations on that.
> - Results over the F functions (see tables 1, 6 and 7), with translation is between runs is not sufficient. I think the approach proposed is ny design translation invariant (like convolutions are) and making some kind of L2O in such context is very likely to let super efficient results like those reported. In fact, random rotations and noise should have been done (like it is with the BBOB, https://numbbo.github.io/workshops/), to ensure that the benchmarks are less skewed toward the invariance of the proposed approach. Even better, we should look at how training the approach on some F functions and tested it on other F functions would have been very interesting, to test the "generalization" capacity of the approach.

---

> > ### Comment · Reviewer_Tk68 · 2022-12-02
> > **Update of my evaluation following responses and paper update (cont.)**
> >
> > - I still maintain that the computational efficiency missed the point that the computational bottleneck of black-box optimization approach is the fitness evaluation for real-life optimization problems, where intense simulations are usually required to evaluate the quality of each candidate solutions. Executing most standard black-box optimization algorithms over simple analytical equations like the F functions is very fast, most algorithms having a complexity of something like O(d m), where d is the dimensionality of x and m is the population size, which are both values in the 10s or 100s each (i.e., very low). CMA-ES scales less well given it has to manage a covariance matrix of size d x d, but is usually also very efficient for problems of dimensionality less than 100. Using a GPU to optimize the algorithm provides no clear advantages in most cases, if fitness evaluation cannot be also done on GPU. Of course, with the current approach being able to execute the convolution on GPU is interesting, but that is to circumvent a limitation of the proposed approach, which is more computationally demanding and most usual black-box optimization algorithms.

---

> > ### Author Response · Authors · 2022-12-03
> > **Response to New Comments (Part 2)**
> >
> > **Q4**: Others, like CMA-ES with its adaptation of a covariance matrix, is able to learn when executed on a given optimization problem, although it is not learning across several runs. Let's compare apples with apples and do not claim that black-box optimization algorithms are not able to do L2O when they are not designed for that.
> >
> > **Answer 4:** We did not state that black-box optimization algorithms are not able to do L2O. We just state that current EAs are not able to do L2O. We hope you can give examples of existing EAs that can do L2O. We think that CMA-ES may not be an L2O framework. The reasons are summarized as follows:
> >
> > 1	Learning to optimize (L2O) is an emerging approach that leverages machine learning to develop optimization methods, aiming at reducing the laborious iterations of hand engineering. It automates the design of an optimization method based on its performance on a set of training problems. We believe that CMA-ES does not fit this definition.
> >
> > 2	We did not find the statement that CMA-ES is an L2O framework in the review literature of L2O [2].
> >
> > Your last sentence shows the side: currently no EAs can do L2O. However, DECN do it.
> >
> > [2] Tianlong Chen, et al. Learning to optimize: A primer and a benchmark. JMLR, 23:1–59, 2022.
> >
> > **Q4**: In my review, I stated that the convolutions are used in replacement of the variation operators (crossover and mutation), which appears to raise a reaction from the authors. I still maintain this is the case, the proposed approach of modeled against the generic EA iterative optimization model (let's call it population-based optimization if the evolutionary term appears shocking or inexact), where a population of candidate solutions is modified (variation operator) and resampled according to their performance (selection) over many iterations (generations) until convergence. The learned convolutions proposed in the paper is simply a way to learn the variation operator.
> >
> > **Answer 5:** We agree with your current version of the statement. Your previous statement caused us to misunderstand.
> >
> > **Q6**: The justification of using convolution and using an L x L structure is weak, the underlying principles relying on convolution are not clear to me.
> >
> > **Answer 6:** Similar to the crossover operator in GA, the purpose of this is to facilitate the information exchange between individuals in the population (See Section 3.2, the first paragraph).
> >
> > **Q7**: Proposing a yet new optimization approach is not very interesting as far as I am concerned, but proposing new, sound principles for black-box optimization would be. I would have liked to get better articulated explanations on that.
> >
> > **Answer 7:** We agree that proposing a new optimization method is not science; how to solve problems is.
> > Our contribution is shown as follows: We propose an L2O framework to automatically learn an optimization strategy for the target task to overcome the poor adaptability of the current EA to the target task.
> >
> > **Q8**: the "generalization" capacity of the approach.
> >
> > **Answer 8:** In the updated version, the part of Results on Low-fidelity Training Dataset (See Section 4.1 and Table 2) tests the DECN's "generalization" ability. We train DECN on F1-F3 functions and test it on F4-F9 functions.
> >
> > **Q9**: Using a GPU to optimize the algorithm provides no clear advantages in most cases if fitness evaluation cannot also be done on GPU.
> >
> > **Answer 9:** We are confused about the discussion on this point. You've been attacking this non-existent argument, which has nothing to do with our article.
> >
> > DECN does not accelerate fitness evaluation. We also do not claim that DECN accelerates fitness evaluation. We are just stating the fact that DECN can use the GPU to accelerate the optimization process. Most importantly, this is not our main contribution. We also do not claim that the EA algorithm is slow and the complexity of DECN is low. We can use GPU to compensate for the running time gap between EA and DECN. Experimental results show that our scheme can indeed bridge this gap even better. **Most importantly, DECN converges faster. DECN can achieve better performance with a small number of function evaluations.**
> >
> > In future work, we will consider how to speed up fitness evaluation for specific tasks.

---

> > ### Author Response · Authors · 2022-12-03
> > **Response to New Comments (Part 1)**
> >
> > Thank you very much for your reply. We have taken your questions seriously. We also welcome your other questions.
> >
> > First of all, we are very sorry that the inaccurate description has caused you trouble.
> >
> > Now that you know EAs well, we assume you understand the pain points of EAs. It is common knowledge that EA has poor adaptability to target tasks. Even CMA-ES cannot escape this curse; otherwise, there would not be so many improved versions of CMA-ES. You keep saying EA is "good enough". So, why is the work on journals/conferences such as IEEE TEVC, GECCO, PPSN, etc., continuing to improve it? Especially for new application scenarios, many efforts focus on redesigning EA to adapt to the task. DECN is a big step forward in improving the adaptability of optimization methods to target tasks. We are delighted to share the latest results with you, and welcome your questions. We believe that ICLR is just a platform to display our results, and whether to publish them is not our ultimate goal. We just want to get everyone's approval and feedback on the mechanism. Since you are an expert in EAs, we sincerely hope you can rationally view DECN's contributions to the evolutionary computation community. Thank you very much.
> >
> >
> > **Q1**: About the argument that "The generalization ability of EAs is poor. Faced with a new black-box optimization task, we need experts to redesign the EA’s crossover, mutation, and selection operations to maximize its performance on the target task, resulting in a hand-designed EA with big application limitation."
> >
> > **Answer 1:** We are very sorry that our clerical error has caused you trouble. "generalization" should be "adaptability".
> >
> > This part is modified as follows: Although the generalization ability of EAs is good, the no-free lunch theorem [1] tells us that we need to specialize an algorithm to a specific problem. Faced with a new black-box task, we need experts to redesign the EA’s operations to maximize its performance on the target task, resulting in poor adaptability.
> >
> > We think this is a very important problem in the field of evolutionary computation.
> >
> > [1] Wolpert, D. H., & Macready, W. G. (1997). No free lunch theorems for optimization. IEEE transactions on evolutionary computation, 1(1), 67-82.
> >
> > **Q2**: I think it is deceiving to claim that generic black-box algorithms are not able to learn or are not generalizing.
> >
> > **Answer 2:** The black-box algorithms contains many types of algorithms. The object of our statement is EAs, not generic black-box algorithms. We dare not make such statements about generic black-box algorithms.
> >
> > We also do not state that black-box algorithms (EAs) are incapable of learning. Our exact words are: Most importantly, due to the limitation of expert knowledge, only little target function information is used to assist the design of EA, which makes it challenging to adapt to the target task. How to automatically design optimization strategies according to new tasks is crucial.
> >
> > We just think that EAs cannot fully use the information of the target task to aid the design of EA. Experimental results also verify the correctness of this statement.
> >
> > **Q3**: Several of them (e.g. genetic algorithm) are not designed to learn but are still meta-heuristics, that can be applied to a wide variety of problems -- they are generalizing, although they are not able to specialize that much to a given context. Of course, some hyper-parameter adjustments may be helpful, like it is with most ML algorithms. Of course, specializing an algorithm to a specific problem may provides some gains at the expense of wide applicability (remember the no free lunch theorem), but that's not something that commonly done when using EAs.
> >
> > **Answer 3:** By these statements, we take you to admit that there is much room for improvement in the performance of EAs on the target tasks. This is a crucial issue. However, you cannot deny that DECN has made great strides in this regard.

---

> > ### Author Response · Authors · 2022-12-11
> > **Did we clarify your comments?**
> >
> > Sorry to bother you. Did our answer clarify your query? We very much welcome your new questions.
> >
> > Best Wishes
> >
> > Authors

---

### Author Response · Authors · 2022-11-14
**General Comments**

We sincerely thank all reviewers for their constructive comments and valuable suggestions. Following their suggestions, we have carefully revised our paper.  Point-to-point responses can be found below to each reviewer. We are also glad to improve our work to address any further concerns continually.

**Motivation**

The adaptability ability of current evolutionary algorithms (EAs) is poor. Faced with a new black-box optimization task, we need experts to redesign/select the EA's crossover, mutation, and selection operations (including their hyperparameters) to maximize its performance on the target task, resulting in a hand-designed EA with significant application limitation. Most importantly, due to the limitation of expert knowledge, only little target function information is used to assist the design of EA, which makes it challenging to adapt to the target task. How to automatically design optimization strategies according to new tasks is crucial. To the best of our knowledge, there is currently no work to address this issue. We think EA is a generative optimization model that realizes the generation from a random population to an optimal solution by manually designing crossover, mutation, and selection operations. These operations aim to generate potential solutions and retain good ones. The task of automatically designing an optimization strategy is learning how to generate and retain potential solutions automatically. This paper is to show how DECN finish this task.

By constructing a set of differentiable surrogate functions of the objective black-box function, DECN can allow the designed CRM and SM to learn the strategy of optimizing the objective function. At this point, DECN effectively utilizes the information of the target black-box function to assist the construction of the optimization strategy. The degree of fit of DECN with the target task is much higher than that of the human-designed EA. The following statement may be one-sided: Bayesian optimization also suffers from poor generalization. For example, how to choose/design appropriate acquisition functions for different problems.

We also found that DECN is a learning-to-optimize framework. Therefore, DECN is compared with the learning-to-optimize framework (L2O-Swarm) for black-box optimization. At the same time, Bayesian optimization is a typical black-box optimization algorithm, so we also compare it with the advanced Dragonfly method.

**Results**

The optimized objective function values are shown in the table. The number in brackets is the standard deviation of repeated experiments.
To verify our motivation, DECN is compared with the optimally designed DE, ES, and CMA-ES. The population sizes of DE, ES, CMA-ES, and DECN are 100. DE, ES, and CMA-ES run for 100 generations. For DECNws3, its architecture determines that DECN has only been iterated for three generations. DE, ES, and CMA-ES have 100/3 times as many function evaluations as DECN, which is highly unfair to DECN. Both Dragonfly and L2O-Swarm run to convergence. However, on high-fidelity training datasets and two real-world problems, DECN outperforms these five algorithms. This result validates the motivation of this paper, and there is a great advantage in automatically learning the optimization strategy.

Especially in Table 1, our scheme outperforms existing methods by a large margin. This is because when training DECN, we use a high-fidelity surrogate function of the target black-box function. The trained DECN contains an optimization strategy that is more tailored to the task. Current DE, ES, CMA-ES, and Dragonfly do not use this information to design algorithms. Even if we constantly adjust the hyperparameters of the comparison algorithm, the results are unlikely to be better than DECN. Of course, this is based on the fact that DECN can indeed effectively use this information to learn optimization strategies. This is what DECN is all about. We also test the performance of DECN trained on the low-fidelity surrogate functions, as shown in Table 2. Compared with these five algorithms, deep DECN still has a significant advantage.

We have uploaded the modified version and the modified part is marked in blue.

Best Wish,

Authors

---

### Decision · Program_Chairs · 2023-01-20

**Decision:**

Reject

**Justification For Why Not Higher Score:**

1. Motivation and justification of using convolution not only as a new way but also as a better way to seek right optimization schemes or learn the structure for a problem is not clear.  There is a lack of clear interpretation of the underling principles.

2. Claim on the computational efficiency over fitness evaluation is not convincingly supported.

3. There are concerns on the baseline performance in the comparative study with respect to existing techniques.

4. The revision made or promised to be made seems to be quite significant. Not all reviewers are comfortable with it.

**Justification For Why Not Lower Score:**

N/A

**Metareview: Summary, Strengths And Weaknesses:**

A deep evolution convolution network (DECN)  framework is proposed in this paper to automate population-based black-box optimization.  Convolutional operators are introduced for reasoning and selection for generating populations to avoid conventional hard-crafted strategies and therefore DECN is claimed to improve generalization capability.  Experiments are carried out on numerous synthetic and real-world datasets and the results seem to be supportive.  There have been a good discussion between the authors and reviewers on how the DECN framework should be approached.  The authors also put up a meticulous rebuttal to clarify some of the raised concerns. However, there are still numerous remaining issues.  First of all, the motivation and justification of using convolution not only as a new way but also as a better way to seek right optimization schemes or learn the structure for a problem is not clear.  There is a lack of clear interpretation of the underling principles. Second, the claim on the computational efficiency over fitness evaluation is not convincingly supported.  Third, there are concerns on the baseline performance in the comparative study with respect to existing techniques.  Last, the revision made or promised to be made seems to be quite significant. Not all reviewers are comfortable with it.   So given its current state, it can not be accepted.